# Transcriptome Analysis of Carbohydrate Metabolism Genes and Molecular Regulation of Sucrose Transport Gene *LoSUT* on the Flowering Process of Developing Oriental Hybrid Lily ‘Sorbonne’ Bulb

**DOI:** 10.3390/ijms21093092

**Published:** 2020-04-27

**Authors:** Jiahui Gu, Zhen Zeng, Yiru Wang, Yingmin Lyu

**Affiliations:** Beijing Key Laboratory of Ornamental Plants Germplasm Innovation & Molecular Breeding, China National Engineering Research Center for Floriculture, Beijing Laboratory of Urban and Rural Ecological Environment, College of Landscape Architecture, Beijing Forestry University, Beijing 100083, China; gujiahuibjfu@163.com (J.G.); zengzh2020@163.com (Z.Z.); Wang1R@163.com (Y.W.)

**Keywords:** Oriental hybrid lily, bulb development, *SUT*, vernalization, flower bud differentiation

## Abstract

The quality of Lily cut flower was determined by the quality of bulbs. During the process of vernalization and flower bud differentiation, sugar massively accumulated in the bulb, which influenced the bulb development. However, the details of sugar genes’ regulation mechanism for these processes were not fully understood. Here, morphological physiology, transcriptomes and gene engineering technology were used to explore this physiological change. Seventy-two genes of 25 kinds of sugar metabolism-related genes were annotated after re-analyzing transcriptome data of Oriental hybrid lily ‘Sorbonne’ bulbs, which were generated on Hiseq Illumina 2000. The results showed that these genes were closely related to lily bulb vernalization and development. Combining gene expression pattern with gene co-expression network, five genes (Contig5669, Contig13319, Contig7715, Contig1420 and Contig87292) were considered to be the most potential signals, and the sucrose transporter gene (*SUT*) was the focus of this study. Carbohydrate transport pathway and genes’ regulation mechanism were inferred through a physiological and molecular test. *SUT* seemed to be the sugar sensor that could sense and regulate sugar concentration, which might have effects on other genes, such as *FT*, *LFY* and so on. *LoSUT2* and *LoSUT4* genes were cloned from Oriental hybrid lily ‘Sorbonne’ by RACE, which was the first time for these genes in Oriental hybrid lily ‘Sorbonne’. The physiological properties of these proteins were analyzed such as hydrophobicity and phosphorylation. In addition, secondary and tertiary structures of proteins were predicted, which indicated the two proteins were membrane proteins. Their cellular locations were verified through positioning the experiment of the fluorescent vector. They were highly expressed in cells around phloem, which illustrated the key role of these genes in sugar transport. Furthermore, transient expression assays showed that overexpressed *LoSUT2* and *LoSUT4 in Arabidopsis thaliana* bloomed significantly earlier than the wild type and the expression of *FT*, *SOC1* and *LFY* were also affected by *LoSUT2* and *LoSUT4,* which indicated that *LoSUT2* and *LoSUT4* may regulate plants flowering time.

## 1. Introduction

Sucrose is the major solute for most plant species. It can not only provide nutrients for plant growth and development as the carbon source, but also participate in the signal transduction process in plants as a signal substance [1,2]. Therefore, modulation of carbohydrate metabolism plays an essential role on plant growth, and it needs to be carefully coordinated with other factors such as environment. Sugar can act as signaling molecules and/or as global regulators of gene expression [3,4]. However, how does sugar participate in the process of sensing? and transmitting signals remains largely unknown.

In general, sucrose is the predominant carbohydrate transported from sink to source tissues for long-distance translocation by sugar transporters, among which sucrose transporters (*SUTs*) play an important role in phloem loading and unloading [5,6]. The first isolated *SUT* gene was from spinach [7]. So far, more than 85 putative *SUT* sequences from at least 35 species of higher plants have been lodged in the NCBI GenBank [8]. The vast majority of the *SUT*s cloned belong to the major facilitator super-family (MFS) [9], which was characterized by 12 predicted plasma membrane-spanning helices arranged in two groups of six, separated by a cytoplasmic loop. Some SUTs are characterized by large cytoplasmic loop regions and are reputed to play a role in sugar sensing. SUTs mainly mediate sucrose transport by H^+^ concentration gradient inside and outside the cell, so it is also known as an H^+^/sucrose co-transporter protein [10]. Many reports have shown the transcriptional regulation of *SUT* genes in various developmental processes, by environmental factors such as light and salt stress, and by endogenous signals such as sugars and hormones [11,12,13]. It is likely that *SUT*s are subject to complex regulation at different levels.

SUTs have been divided into five independent clades (SUT1-SUT5) in all plants according to the phylogenetic analysis [9]. The SUT1 clade, which are dicot-specific SUTs, is a kind of high affinity and low capacity (HALC) sucrose carrier and plays the role of collecting low concentration extracellular sucrose from vascular tissues during phloem loading and long-distance transportation [14]. Clade SUT2 and SUT4 are common to both monocotyledons and dicotyledons, but SUT2 is located on the cell membrane, while SUT4 subcellular localization shows that it is located on the vacuolar membrane [15]. Clade SUT3 and SUT5 are monocot-specific SUTs and SUT3 is a direct homologous subfamily of SUT5 [9], which has not been characterized in detail.

Lily (*Lilium* spp.) is one of the most important cut flowers in the world. Lily Oriental Hybrids are highly favored by consumers for their beautiful shapes and various colors, such as Sorbonne. As an important reproductive organ of lily, the bulb is crucial to the development of lily cut flower industry. It has been proved that carbohydrate metabolism is closely related to the bulb and shoot growth in the *Lilium* genus [16,17]. However, the past research mostly focused on temperature, photoperiod and hormones. Seldom have we paid attention to sugar, especially to sugar signaling and we know little about the molecular mechanisms that regulate flowering process. For most dicot species, two or more *SUT* genes have been reported and their genetic structures and sucrose transport activity have been well studied. For example, nine *SUT* genes were found in the genome of *Arabidopsis thaliana* and their sucrose transport function was verified by the expression of yeast cells [15]. *SUT1*, *SUT2* and *SUT4* genes were cloned in tomato, potato and *Pyrus*. [18,19]. In contrast to numerous studies on sucrose transport in dicotyledons, the function of *SUT* genes of many monocotyledons remains largely unknown.

Vernalization is an important step of bulb renewal, and the change of carbohydrate is a key link in vernalization. At present, most studies on the mechanism of lily bulb vernalization have focused on temperature, light, hormones etc., and there is less focus on the metabolism of carbohydrate in the process, even if they are mainly concentrated on physiology like determination of sugar content, while the molecular mechanism needs to be further studied. In addition, it has been reported that carbohydrate metabolism genes have an impact on the processes of plant flowering [3], but the research in lily is still a blank state. The aim of this study was to find the key carbohydrate metabolism genes in the development of lily bulbs. Through the study of Oriental hybrid lily ‘Sorbonne’, we further explored the importance of carbohydrates in the development of oriental lily bulbs, and the internal molecular regulation mechanism of the carbohydrate metabolism gene (*SUT*). Finally, bioinformatics analysis and transgenic *Arabidopsis thaliana* were used to verify the effect of *SUT* gene on plant flowering and to provide more information and evidence for the molecular regulation mechanism in the stage of lily bulb development.

## 2. Results

### 2.1. Analysis of Carbohydrate Metabolism Gene Annotation

In our previous research, we studied the molecular mechanism of vernalization and flower formation by analyzing transcriptome data of ‘Sorbonne’ [20]. In that research, exploration of flower differentiation-related genes was the focus and the information of carbohydrate metabolism that was related was not fully analyzed. Therefore, in this study, the origin transcriptome data were re-analyzed and re-annotated in the hope of further exploration the relationship between carbohydrate metabolism genes and flowering genes. We annotated all the contig sequences of carbohydrate metabolism-related genes in this transcriptome data. Taking the Invertase gene (INV) as an example, the gene sequences and protein sequences were collected in the nucleotide database of NCBI. The collected gene sequences were compared with the results of the transcriptions in the transcripts of Oriental hybrid lily ‘Sorbonne’, and the number of *INV* genes was counted. As shown in Table 1, Contig87326, Contig31833, Contig28979, Contig84458, and Contig43251, which do not meet the requirements, were eliminated. While Contig88348, Contig28073, and Contig8633, which basically met the requirements, were kept. Contig1581, Contig11809, Contig6425, Contig17492, and Contig16491 were fully compliant and were reserved.

Finally, 70 sequences related to carbohydrate metabolism with reliable annotations were obtained (Appendix A). To explore the role of carbohydrate metabolism genes further, we analyzed their expression in different stages of bulb development. Results showed that more than half of the genes in the carbohydrate metabolites were active in the vernalization and flower bud differentiation period of the apical meristem of oriental lily ‘Sorbonne’ (Figure 1), which indicated that they were actively involved in the lily flower bud differentiation process, but its detailed ways of participation remained to be explored. The most active genes were genes of the Fructose-1,6-Bisphosphata Aldolase (FBA) gene family including Contig3235 and Contig4117; genes of Sucrose Synthase (SS) gene family including Contig17518, Contig9581, Contig6434, and First-Contig10; and Contig1555 which belonged to Hexokinase (HXK) gene family

One other point that really caught our attention was that even with genes of the same family, the expression patterns of the same time were not the same, or even extremely different. For example, Contig1555 and Contig46868 belonged to the *HXK* gene family, but Contig1555 was significantly expressed at each period and Contig46868 was almost completely opposite. Similarly, the expression of Contig17518, which belonged to the *SS* gene family, was much higher than Contig87292. A similar phenomenon occurred in Sucrose Will Eventually be Exported Transporters (SWEET) gene family and Terpene Synthase (TPS) gene family, which suggested the complexity and rigorous gene expression (Figure 2).

### 2.2. Analysis of Gene Expression Patterns of Carbohydrate Metabolism

In order to further understand the expression regularity of carbohydrate metabolism genes in periods of Sor25, Sor4 and LXH of oriental lily ‘Sorbonne’, this study clustered the above genes according to the RPKM changes in the three periods. Results showed that they were divided into four kinds of expression patterns including continuous up-regulation (Cluster 1), first rise and then fall (Cluster 2), first drop and then rise (Cluster 3), and continued to decline (Cluster 4).

The expression pattern of most carbohydrate metabolism genes belonged to Cluster 3, that was, the expression was reduced in the vernalization period and restored after the vernalization. It is presumed that the gene may be inhibited by low temperature and not participate in the vernalization of the apical meristem, but it is involved in late flower bud differentiation. The majority of these genes, including Contig3235, Contig4117, Contig17518, Contig9581, and Contig6434, were at high expression levels (Figure 1), which indicated they actively participated in the life activities of lily apical meristem and played the role of energy supply rather than as a key gene or signal to regulate the vernalization and growth process of lily bulbs.

The expression of genes in Cluster 4 was continuously reducing, suggesting that it may be independent of vernalization and differentiation of lily, and may even be an inhibitory factor. Among them, though, the expression of *HXK* family gene Contig1555 continued to decreased, and still remained at a relatively high level (Figure 1), indicating that it was a main role in the sugar supply rather than a vernalization inhibitory factor. Gene-relative expression of Contig50329, which also belonged to the HXK family, was negative, showing that it may be the inhibitory factor of vernalization and differentiation of lily.

During the vernalization, the gene expressions of Cluster 1 and Cluster 2 were both increased and were not inhibited by low temperature, showing that these genes could promote the apical meristem to break the dormancy and complete the vernalization. However, Cluster 1 genes increased more slowly than Cluster 2 genes. After the end of the vernalization period, the expression of Cluster 1 genes continued to rise to the period of flower bud differentiation, indicating that genes of Cluster 1 were further involved in flower bud differentiation, and may be the flower-promoting genes. The expression of Cluster 2 genes decreased slightly during flower bud differentiation, suggesting that it could be mainly involved in promoting the vernalization process of apical meristem, and it was no longer the key gene during the period of flower bud differentiation. Contig1420 of Tonoplast Monosaccharide Transporter (TMT) family gene, Contig5669 of *SUT* family gene, Contig13319, and Contig8288 of Glucose Transporter (GLUT) family gene, were the Cluster 2 genes, and all of them belonged to the sugar carrier genes involved in the active transport of sugar in phloem and cells inside to outside to regulate cells osmotic pressure. The number of Cluster 1 genes was the least, and half of them were *SS* family genes, such as Contig1205, Contig9247 and first-Contig10, which were located downstream of the *SUT* gene and were directly regulated by the *SUT* gene. According to the two expression patterns, it is possible that Cluster 2 genes may be in the upstream of Cluster 1 genes, and Cluster 1 genes may be the key factor for lily to break dormancy.

### 2.3. Analysis of Co-Expression Patterns of Genes

Gene co-expression network (GCN) is a kind of molecular biological method where microarrays or deep-sequencing (RNA-seq) were used to judge the interactions of genes. Genes are usually represented as nodes, and edges represent pairwise relationships between genes. In order to explore the correlation between carbohydrate metabolism genes and genes related to flower differentiation, Auxin, Gibberellin, cold and methylation, this study collected reliable sequences of the above genes with significant differences of expression in the transcriptome data [20] to establish Carbohydrate Metabolism Gene Co-expression Patterns. When setting *p* = 0.95 (Pearson’s correlation coefficient) for the standard selection of the correlation coefficient, the carbohydrate metabolism genes were closely related to all other genes. It was further stated that carbohydrates were not only energy-supplying substances, but played a pivotal role in the regulation of gene expressions. The carbohydrate metabolism genes were extremely close to flower differentiation, auxin and methylation genes, and these kinds of genes may co-ordinate the impact of the process in lily apical meristem. However, it is difficult to distinguish the importance among them because almost all kinds of genes were related. The correlation coefficient of the standard selection was then improved to 0.99. The number of genes and associated lines were greatly reduced. There were 30 carbohydrate metabolism genes that had a collective effect with plant hormones, methylation and other related genes, and some of these carbohydrate metabolism genes may be the key genes to impact vernalization and the flower bud differentiation process of lily (Figure 3).

### 2.4. Changes in Carbohydrate and Hormone Content

Since carbohydrate metabolism genes were associated with other genes closely, to further investigate the role of carbohydrate in the development of lily, the change of carbohydrate and hormone content in inner and outer scales at different stages of bulb development and flower bud differentiation was measured. Lily bulbs, as important underground storage organs, are rich in carbohydrate, among which starch and sucrose are the main forms [21]. It was shown that starch was a major component of bulbs before venalization; its content could be as high as 300 mg/g (Figure 4). However, with the beginning of vernalization at a low temperature of 4 °C, starch content began to degrade. The decrease rate was the most significant in the early period of vernalization and the middle of vernalization starch content in bulbs reduced by half. After, the starch content continued to decrease, the amplitude became gentle, especially in the flower bud differentiation and development periods. Overall, the decrease of starch content in lily scales was higher than that of inner scales. From the end of vernalization, the starch content of inner scales began to be higher than that of outer scales.

In contrast to the trend of starch content, the content of soluble sugar in the bulb increased significantly, especially during vernalization. At the end of vernalization, the soluble sugar content of lily bulb was more than doubled in the period before vernalization, reaching the maximum. During the flower bud differentiation and development period, the content of soluble sugar decreased slightly, and the changing trend of lily inner and outer scales was consistent. The content of reducing sugar in the Oriental hybrid lily ‘Sorbonne’ changed slightly, became lower at the middle of the vernalization, and was the highest at the end of vernalization. The trend of the inner and outer scales was consistent, and the peak difference was only 2.04 mg/g and 3.04 mg/g, respectively. The trend of sucrose content in lily bulbs was very similar to that of soluble sugar, which was significantly increased during the vernalization period, and the flower bud differentiation and development period decreased slightly, and peaked at the end of vernalization (Figure 4). The sucrose content of the outer scale was higher than that of the inner scale; the peak values were 49.49 mg/g and 39.69 mg/g, respectively.

In the process of vernalization in plants, in addition to the need for carbohydrates to participate in metabolic activities, the content of a signaling substance like plant hormones would also change with plant growth state, and both of them work together in this process. Usually in the plant vernalization period, the growth inhibitory hormones would gradually reduce [22]. ABA content in different periods of lily bulbs changed significantly. The ABA content of the inner scale increased in the early period of vernalization, reached a peak value of 163.61 ng/g in the middle of vernalization, and then decreased significantly to 98.16 ng/g in late vernalization (Figure 4). The ABA content of the outer scales increased steadily with the vernalization and growth of bulbs, indicating that ABA was related to vernalization.

The contents of IAA in both inner and outer scales were decreased at the beginning of vernalization to 55.72 ng/g and 44.38 ng/g, and then increased gradually. The peak reached 144.38 ng/g and 121.39 ng/g, respectively, in the period of flower bud differentiation, and then remained stable. Among them, the content of IAA in the inner layer was significantly lower than that of the outer scale, but the change trend was basically the same. Since the end of vernalization, IAA content gradually increased to promote the development of flower bud.

The GA3 content of lily bulbs in different times was much lower than that of IAA and ABA and the variation trend was more gentle. However, there was a significant difference between inner and outer scales. The content of GA3 in the inner layer increased significantly at the beginning of vernalization, reached the first peak of 7.86 ng/g in the middle of vernalization, and then decreased sharply during the flower bud differentiation stage, reaching another peak of 7.81 ng/g again at the flower bud development stage. The content of GA3 in outer scales was almost not changed.

Plant vernalization is the result of a variety of auxin effects. In order to explore their interactions, this study focused on their relative changes in content (Figure 4). The ratio of IAA to ABA in the bulb during vernalization decreased first and then increased, and the trend was further improved after the dormancy was released. The results showed that the content of IAA decreased during the middle period, which was beneficial to the lily vernalization, and then increased gradually to promote the differentiation development of lily. The ratio of GA3 to ABA of outer scales continued to increase steadily while the inner scales increased significantly during the period of flower bud development (Figure 4), indicating that GA3 may play an important role in the development of flower bud.

### 2.5. Analysis of SUT Gene and Carbohydrate Metabolism Genes, Flower Differentiation Genes and Hormone Genes Expression

Sucrose is the main form of soluble sugars transport and there are two methods to transport sucrose into the tissue cells: one is through the intercellular connection (PDS), and the other one is by the sugar transporter. Among them, the sucrose transporter is a key path for sugar transportation (Figure 5) [9]. According to analysis of transcriptome data, *SUT* gene was found to play a key role in the process of vernalizaiton and was closely related with flower differentiation genes and hormone genes.

The expression of SUT and carbohydrate metabolism genes in vernalization and flower bud differentiation is shown (Figure 6). The expression of the *SUT* gene in vernalization had little fluctuations. In addition, its expression, compared with other carbohydrate metabolism genes, was at higher levels, especially in early vernalization. The expression of *TMT* gene was kept at a high value, and the expression trend was earlier than SUT. The expression of *VINV* gene was significantly inhibited by low temperature at the beginning of vernalization, gradually recovered in the late vernalization, and remained stable in the flower bud differentiation period. The expression trends of *HXT* and *SUT* genes were similar in the vernalization period, which indicated that both of them played a role in intracellular sugar accumulation. The expression of the *SS* gene period increased steadily with the accumulation of carbohydrate before vernalization and increased with the change of the *SUT* gene in the flower bud differentiation period, which corresponded to the expression pattern in *2.2*.

In addition, we selected the key genes of flower differentiation in the gene co-expression network with SUT in *2.3*, and the expression patterns are shown in Figure 6. The expressions pattern of Flowering Locus T (FT) gene was similar to that of Floricaula (LFY) gene, which was inhibited by low temperature in the early stage of vernalization, recovered in the later period, and reached the peak at the stage of flower bud differentiation. The overall expression trend was lagging behind that of SUT, but the expression pattern was similar. The expression of both Vernalization (VRN) gene and Suppressor of Overexpression of Constant1 (SOC1) gene reached a significant peak value at the end of vernalization and had no obvious connection with *SUT* gene.

The expressions of *SUT* and hormone-related gene are shown in Figure 6. IAA-related gene was obviously inhibited by low temperature at the beginning of vernalization, gradually recovered at the late vernalization, and reached the peak in the flower bud differentiation period. *DELLA* gene was significantly expressed in the early stage of vernalization and shared a similar expression trend with the IAA-related gene, which first increased and then declined. ABA-related gene was also highly expressed at the beginning of vernalization but decreased sharply in late vernalization and then rose again during the flower bud development.

### 2.6. Full Length Cloning of Genes

The Physicochemical properties of the proteins of LoSUT2 and LoSUT4 were analyzed, and the secondary structures of these proteins were predicted. The total length of the coding region of LoSUT2 mRNA is 1494 bases (from the start codon to the stop codon), encoding 497 amino acids. The chemical formula of the protein is C2436H3834N640O644S21; the relative molecular mass is 53064.4; and the theoretical isoelectric point pI is 9.30. There existed 23 negatively charged amino acid residues (Asp + Glu) and 33 positively charged amino acid residues (Arg + Lys); the protein may be positively charged. The instability coefficient is 33.52, which is considered to be a stable protein. When assuming that all cysteines were thought to constitute Cystine, the extinction coefficient and absorbance (protein concentration is 1%, or 1 g/L) were 84,380 (M^−1^ cm^−1^) and 1.590, respectively. When all cysteines were assumed to be present alone, the extinction coefficient and absorbance were 83,880 (M^−1^ cm^−1^) and 1.581, respectively. In the LoSUT2 protein sequence, the proportion of each amino acid was Leu 13.1%, Ala 11.7%, Gly 9.9%, Val 8.2%, Ile 7.0%, Ser 7.0%, Pro 5.6%, Arg 5.4%, Thr 5.4%, Phe 4.6%, Asn 2.8%, Gln 2.4%, Glu 2.4%, Met 2.4%, Trp 2.4%, Tyr 2.4%, Asp 2.2%, Cys 1.8%, His 1.8%, and Lys 1.2%. The proportions of hydrophobic amino acids and hydrophilic amino acids were 48.8% and 51.2%, respectively, and the hydrophilic average coefficient was 0.62 (Figure 7).

The total length of the coding region of LoSUT4 mRNA is 1733 bases (from the start codon to the stop codon), encoding 590 amino acids. The chemical formula of the protein is C2865H4438N762O805S31, the relative molecular mass is 63431.2; the theoretical isoelectric point pI is 7.16. There were 38 negatively charged amino acid residues (Asp + Glu), and also 38 negatively charged amino acid residues (Arg + Lys). The protein may not be charged. The instability coefficient is 36.19, which is considered to be a stable protein. The total extinction coefficient and absorbance (protein concentration 1%, or 1 g/L) were 87,150 (M^−1^ cm^−1^) and 1.374, respectively, assuming that all cysteine was thought to form cystine. The extinction coefficient and absorbance were assumed to be 86,400 (M^−1^ cm^−1^) and 1.362, respectively, when all cysteines were present. In the LoSUT4 protein sequences, the proportion of each amino acid was Gly 10.5%, Leu 10.3%, Ser 9.3%, Val8.8%, Ala 8.6%, Phe 6.4%, Thr5.4%, Ile 5.3%, Arg 4.7% Pro 4.2%, Asn 4.1%, Asp 3.2%, Glu 3.2%, Met 3.2%, Gln 2.9%, Trp 2.2%, Cys 2.0%, His 2.0%, Lys 1.7%, and Tyr 1.7%. The proportions of hydrophobic amino acids and hydrophilic amino acids were 44.7% and 55.3%, respectively, and the hydrophilic average coefficient was 0.370 (Figure 8).

LoSUT2 contained 69.6% spiral regions, 3.4% extension chains, and 27.0% cycle areas. In terms of solvent accessibility, 81.49% of the sequence was included inside the cell membrane, 17.30% of the sequence was exposed outside the cell membrane, and 1.12% of the sequence was intermediate. No possible signal peptide or acetylation was predicted. It has been reported that SUT protein can be phosphorylated [23], so it is necessary to predict the possible phosphorylation sites of LoSUT2, and finally we predicted that 8 Ser, 8 Thr and 1 Tyr may be phosphorylated (Figure 8).

LoSUT4 contained 58.5% spiral regions, 4.9% extension chains and 36.6% cycle areas. In terms of solvent accessibility, 70.85% of the sequence was included inside the cell membrane, and 29.15% of the sequence was exposed outside the cell membrane. No possible signal peptide or acetylation was predicted. It was predicted that 20 Ser, 3 Thr and 4 Tyr may be phosphorylated (Figure 8).

As a membrane protein, it is necessary to understand the transmembrane mechanism of LoSUT2 and LoSUT4. We used Phobius to predict the transmembrane structure of the protein (Figure 9). Both proteins were predicted to have 12 transmembrane regions, and it was concluded that all SUT genes belonged to a large population of sucrose carriers, and their typical characteristics were 12 transmembrane regions [9,10]. The transmembrane region was divided into two groups based on the middle of a circular region located in the cell. There were six transmembrane regions in each group. The N-terminal and C-terminal were all inside the cell, which was consistent with LoSUT2 and LoSUT4. It was also indicated that the SUT protein had a conserved histidine N in the first extracellular loop region, which was closely related to the transport activity of sucrose. In addition, in *LoSUT2* it was also predicted that there was a histidine residue in the first extracellular loop region (44–62 sequence, SLLTPYVQELGIPHKWSS), and the same as *LoSUT4* (76–94 sequence: LSLLTPYVQTLGIGHFFSS), which further supported both genes that belonged to the SUT family.

### 2.7. Evolutionary Tree

It is important to determine the evolutionary status of LoSUT2 and LoSUT4 in the SUT gene. Therefore, we collected the DNA sequences of 86 *SUT* genes, including 64 *SUT* genes of 28 dicotyledonous plants, 16 *SUT* genes of 7 monocotyledonous plants, and 6 *SUT* genes of bryophytes to build an evolutionary tree with LoSUT2 and LoSUT4.

All *SUT* genes are broadly divided into three sorts. The gene of Type-I SUT contains the dicot transporter of the classical SUT1 of dicotyledonous plants, which is thought to play a role in the phloem loading. Type-II includes all *SUT1* genes of monocotyledonous plants, such as *OsSUT3,4,5* and genes of the SUT2 family of dicotyledonous plants. Among them, parts of the SUT2 family of genes are presumed to be sugar receptors. Type-III contains the majority of the *SUT4* genes in the dicotyledons and part of the *SUT2* gene in the monocotyledonous plants [15].

LoSUT2 belonged to the Type-III SUT gene and, at the same time, belonged to the same branch with *SUT2* genes, including OsSUT2, ZmSUT2 and HvSUT2 of the monocotyledonous plant. LoSUT4 was part of the *SUT* genes of Type-II and belonged to the same branch with *SUT4* genes like OsSUT4, ZmSUT4 and AcSUT4 (Figure 10). It was further explored that the homology of LoSUT2 with OsSUT2, ZmSUT2 and HvSUT2 was 69.23%, 69.33% and 69.29%, respectively. The homology of LoSUT4 and AcSUT4, ZmSUT4 and OsSUT4 was 65.0%, 67.6% and 67.6% respectively.

Several *SUT* genes, including the majority of the monocotyledons of the *SUT* genes, were selected from the phylogenetic tree, and the protein sequences of LoSUT2 and LoSUT4 were compared together (Figure 11). The *SUT* gene is characterized by non-conservative sequences at both ends and the middle, especially the dicotyledonous *SUT2* gene and *LoSUT4*, *AcSUT4*, *ZmSUT4*, and *OsSUT4* in type-II; and the *SUT4* genes of these four monocotyledons have a long non-conservative area. In contrast, the monocotyledonous *SUT1* gene in type-II; the dicotyledonous *SUT* gene of type-I; and the monocotyledonous *SUT2* of type-III, including the *LoSUT2* gene and the dicotyledon *SUT4* gene, are relatively short (Figure 11). In addition, when the transmembrane regions of *LoSUT2* and *LoSUT4* were compared with the multiple sequences of proteins, it was found that the sequences of the transmembrane regions were fairly conserved and the intracellular and extracellular sequences were relatively non-conservative. The sequences of these cells inside and outside may infect LoSUT identification and binding of the substrate, thus varying widely in different species due to identification and binding of the substrate.

### 2.8. 3D Structural Models of Protein LoSUT2 and LoSUT4 and its Subcellular Localization in Onion Epidermis

The images show details of three-dimensional structural models of protein *LoSUT2* and *LoSUT4* (Figure 12). There existed a barrel-like channel surrounded by spiral areas, which may serve as a way to transport sucrose. In order to clarify the subcellular localization of *SUT* gene, Green Fluorescent Protein (GFP) was added to the LoSUT2-pBI121- and LoSUT4-pBI121-expressing vectors. One of the most useful features of GFP is its ability to be used as an in vivo marker in real time without causing any perceptible tissue damage. Positive transformants were identified by highly visible punctate green fluorescence when viewed by epifluorescent microscopy. As shown in Figure 13, GFP-tagged sucrose transporter was located in the cell membrane.

### 2.9. Phenotypic Changes and Fluorescence Quantitative Analysis of Transgenic Arabidopsis thaliana

To explore the function of *LoSUT* in signaling, transgenic *Arabidopsis thaliana* plants overexpressing *LoSUT2* and *LoSUT4* were generated. It was found that Columbia wild-type was able to grow and breed normally with *Agrobacterium tumefaciens* containing LoSUT2 and LoSUT4 recombinants by the inflorescence dip method. After the resistance screening and PCR identification, the over-expressed LoSUT2 and LoSUT4 transgenic plants were obtained (Appendix A). In this study, wild-type (WT) plants and transgenic plants expressing LoSUT2 and LoSUT4 were cultured under the same conditions. The results displayed that the transgenic plants over-expressing LoSUT2 and LoSUT4 showed certain differences comparing with WT plants. By observing its phenotypic changes, it was found that transgenic plants over-expressing LoSUT2 and LoSUT4 flowered ahead of time compared with WT plants (Figure 14).

In order to further verify the changes in the expression of *SUT* gene from the molecular level and to deduce the gene functions of LoSUT2 and LoSUT4, the WT plants and LoSUT2 and LoSUT4 transgenic plants were analyzed by fluorescence quantitative analysis. As shown in Figure 14, it was obvious that the expression of the *SUT* gene in transgenic plants was much higher than that of WT plants, especially in the flower tissues. The expression of *SUT* gene in the whole plant was widely distributed and its expression was from high to the low order of stem> leaf> flower> root. As the expression of *SUT* gene in LoSUT2 transgenic plants was significantly increased, the expression of *FT* gene was also up-regulated. While the expression of *SOC1* gene was down-regulated compared with WT plants, and there were no significant changes in *LFY* genes. The expression of *SUT* gene in LoSUT4 transgenic plants was significantly increased, as well as the expression of *FT* and *LFY* genes. While the expression of *SOC1* gene was down-regulated compared to WT plants.

## 3. Discussion

### 3.1. Effects of SUT Genes on Bulb Development of Lily Together with Carbohydrate Metabolism Genes, Flower Genes and Hormone Genes

During the vernalization of lily bulbs, the starch content was apparently reduced, while the content of soluble sugar and sucrose were significantly increased. At that time, lily bulb scales gradually lost water and shrunk, especially the outer scales. It was indicated that during the lily vernalization period, the main material starch was translated into soluble saccharides, which means the phenomenon called ‘low temperature saccharification’ appeared [24], and bulbs act as the source to provided energy for the whole plant’s life activities [25,26]. At the end of vernalization, the content of soluble sugar, reducing sugar and sucrose reached a peak, indicating that the bulbs had broken dormancy and vernalization had finished [27]. Researchers have discovered that sugars, especially sucrose, may not only be energy substances, but also a signal to regulate plant growth and development [1]. We found that soluble sugars, especially sucrose, reached a peak after lily vernalization, indicating that it might be a critical signal.

The expression of *SUT* and carbohydrate metabolism genes in vernalization is shown in Figure 3, Figure 4, Figure 5, Figure 6, Figure 7, Figure 8, Figure 9, Figure 10, Figure 11, Figure 12, Figure 13 and Figure 14 During the vernalization, the expression of SUT had little fluctuations and maintained was at high levels compared with other carbohydrate metabolism genes, especially in the early vernalization. *TMT* is the upstream gene of *SUT* and is responsible for sugar loading and emission in phloem [28]. The expression of the *TMT* gene was high all the time and the expression trend was one period earlier than that of *SUT* gene, indicating that the *TMT* gene may participate in the regulation of *SUT* gene. *VINV*, an important gene for respiration, catalyzing the hydrolysis of sucrose into glucose and fructose (this process is irreversible), is located in the mitochondria [3], and was significantly inhibited by low temperature at the beginning of vernalization. After that, it gradually recovered during the late vernalization and remained stable in the differentiation period. *HXT* and *SUT* were both key genes in the process of transmembrane transport [29], and their expressions were similar. The expression of *HXT* and *SUT* were stable at a high level in the vernalization period, indicating that both of them played a role in intracellular sugar accumulation. *SS* is a downstream gene of *SUT*, acting on sucrose decomposition. During the vernalization period, the expressions changed after the *SUT* genes.

Based on the qPCR study of bulbs carbohydrate and carbohydrate metabolism genes, it was speculated that the changes of sugar concentration in lily bulb lead to the changes of osmotic pressure, thus affecting the expressions of *TMT*, and the growth material was transferred from the bulb scales to the growth point through the phloem, which would affect the expressions of *SUT* and *HXT*. The function of intracellular soluble sugar can be divided into two parts, one part was involved in respiration and the other part was accumulated in the cell and broken down by the *SS* gene [30]. The expression of *SS* was sharply affected by *SUT* during vernalization, while the expression of *INV* remained stable during the differentiation and developmental period, indicating that *INV* was mainly involved in respiration and had no significant association with lily vernalization. *INV* is also a large family [31], and studies have suggested that it may be related to plant resistance to drought-resistant signaling [32,33], but in this study, the *INV*, which has a co-expression relationship with the *SUT*, does not show the signal function.

The most noteworthy point is that elevated intracellular sugar levels inhibited the expressions of *SUT*, but did not affect the expressions of *HXT*. As is shown in Figure 6, after vernalization, the expression of *SUT* was significantly lower than that of *HXT*. In addition, during the development of flower buds, the expression pf *HXT* continued to increase while the expression of *SUT* began to decrease seriously. This regulatory mechanism may be to maintain the balance of osmotic pressure inside and outside the cell. It has suggested that *HXT* genes may have a sugar sensation function [34], but HXT with a co-expression relationship with *SUT* in this study did not show a potent function, which indicated that *INV*, *HXT* signal path and *SUT* signal path were independent.

The expression patterns of *FT1* and *VRN* genes were similar to that of *SUT*, which may be affected by *SUT*. The expressions of *SOC1* and *VRN* genes were not significantly correlated with the *SUT* gene. According to the analysis of sugar physiological results, the change of sugar concentration at the end of vernalization is also a critical find. It is speculated that *SUT* may regulate the expression of flowering genes by influencing sugar concentration, but still needs further verification. Sucrose may be an important signal for flower induction. The additional exogenous sucrose could induce early flowering of *Arabidopsis thaliana* under low light conditions [9]. Compared with normal flowering plants, a large accumulation of sucrose in the apical meristem was detected in early flowering plants. Through the transgenic maize overexpression of *SUT*, it was found that the *SUT* gene was beneficial to maize development [35]. The upregulation or downregulation of *SUT* gene in potatoes also had a significant effect on the flowering time of plants [36].

Carbohydrates are not only essential energy for plant growth and development, but also material that regulate specific growth processes with plant hormone signaling networks, such as embryo formation, seed germination, seedling growth, and bulbs formation [3,37]. In this study, we found that the expression patterns of *SU*T genes were different from that of hormone genes, but *SUT* could affect the expression of a hormone gene by affecting the sugar concentration. Studies have shown that exogenous sugars can promote wild Arabidopsis germination while suppressing ABA concentrations [38]. The effects of different plants were different towards the effects of GA and IAA on flowering. For example, foreign aid GA3 in vitro could make Western azaleas bloom ahead of time [39]. A high level of GA was beneficial to alfalfa formation, while a high level of IAA was not conducive to alfalfa flower [40]. The content of IAA in bamboo flowering late was lower than that of bamboo flowering early [41]. In this study, we found that IAA and DELLA contents were significantly increased and ABA content decreased during the development of lily differentiation. It was speculated that IAA was the promoting factor and ABA and GA3 were negative regulators in lily development. At this time, *SUT* expressions were consistent with the IAA and DELLA, and opposite to ABA, so *SUT* may be a flower growth factor.

### 3.2. The SUT Full Length Sequences and its Genes Function in Oriental Lily

The results of the physical and chemical properties of the *LoSUT* proteins were consistent with the properties of a membrane proteins, which also helps to identify and purify the LoSUT proteins. LoSUT2 and LoSUT4 proteins did not show the presence of signal peptides, suggesting that it was not a secreted protein. This finding was consisted with its location and function as a membrane protein which is responsible for transport of sucrose into the cell of membrane [42]. The results of the localization experiment also confirmed that SUT were membrane proteins. The possible acetylation sites were not predicted on the LoSUT2 and LoSUT4 proteins, but 17 and 27 possible phosphorylation sites were predicted respectively, indicating that the activity of LoSUT2 and LoSUT4 may be regulated by phosphorylation [9]. The high degree of consistency between LoSUT2, LoSUT4 and the classical *SUT* gene on the secondary structure also shows that our analytical results were believable.

*LoSUT2* belongs to the Type-II *SUT* gene and shares the same branch with *OsSUT2*, *ZmSUT2* and *HvSUT2* in monocotyledons. LoSUT4 belongs to Type-III and is closely related to the *SUT4* gene of *OsSUT4*, ZmSUT4 and AcSUT4 of pineapple. This is one of the reasons why we named them *LoSUT*. It is difficult to study the *SUT* genes in monocotyledonous plants, so the known monocotyledonous *SUT* genes are much less than that of dicotyledons [43]. However, we found that the *SUT* genes in the monocotyledonous plants of lily for the first time filled the gaps in the *SUT* genes in the lily, which laid the foundation for the further study of the *SUT* gene functions of the lily.

Studies on transgenic *Arabidopsis* plants showed that the *LoSUT2* and *LoSUT4* genes of lily were located in the cell membrane and were significantly expressed in the cells near the phloem, indicating that they were the key genes in the sugar transport process [44]. Transgenic *Arabidopsis thaliana* plants grew faster than WT plants, and the transgenic plants flowered ahead of time. In the short day, three *Arabidopsis* mutants (*AtSUC9* sucrose transfer genes) were bloomed earlier than the control, and when the gene was damaged, the mutant flowering was later than the control. It was hypothesized that in the *AtSUC9* mutations, *AtSUC9* depressed flowering by reducing sucrose concentration [45], suggesting that *AtSUC9* expression was associated with carbohydrate levels and translations and affected flowering.

Fluorescence quantitative analysis of the transgenic plants revealed that the overexpression of *LoSUT2* and *LoSUT4* genes up-regulated the flowering gene *FT*, which promoted the flowering of plants, indicating that *LoSUT2* and *LoSUT4* genes in lily were flowering promoters. It has been found that sucrose may act directly on the *FT* gene and directly on the *FT* genes [3]. The overexpression of transgenic plants resulted in down-regulation of the *SOC1* gene, which may be due to the high sugar content of the plant. Previous studies have found that sucrose content was too high to delay flowering and reduce *FT* and *SOC1* mRNA transcription levels [46].

The effects of *LoSUT2* and *LoSUT4* genes on the flowering genes were also different by transgenic *Arabidopsis thaliana* qRT-PCR. Among them, the up-regulation effect of *LoSUT2* on the *FT* gene was better than that of *LoSUT4*, and the inhibition of *SOC1* gene was stronger than that of *LoSUT2* gene, which indicated that *LoSUT2* gene had a stronger function in promoting sucrose content than that of *LoSUT4* gene. In addition, the *LoSUT4* gene had a slight effect on *LFY*, while *LoSUT2* did not.

The main reason for the difference in function of the two genes is the difference in their species and in protein constructs. SUT protein can be phosphorylated [9]. LoSUT2 was predicted to have only 17 possible phosphorylation sites, while LoSUT4 was predicted to have 27 possible phosphorylation sites and mostly they were concentrated in the initial region that affected its activity, suggesting that LoSUT4 may be more susceptible to external factors.

*LoSUT2* and *LoSUT4* both belong to the MFS family. However, LoSUT2 belongs to Type-III, which has a close genetic relationship with the *SUT2* gene of monocotyledons. LoSUT4 belongs to type-II and the branch with the *SUT4* genes of monocotyledons, so their functions should be similar. Some genes in Type-II were presumed to be sugar receptors [42], indicating that LoSUT4 may have a function of a sugar receptor. From the structural analysis, only 17.30% of the LoSUT2 sequences were exposed out of the cell membrane and LoSUT4 had 29.29.15% of sequences exposed out of the cell membrane. The main difference of them is located on the 6–7 cycle structure of the protein. LoSUT4 is more in-depth in this region than LoSUT2 cytoplasmic internal. The protein structure of monocotyledonous plant SUT4 is about 40 amino acids more in the middle part than that of monocotyledonous plant SUT2 protein, which may be the key part of the sensor structure, and it can be seen from the protein sequence alignment [43].

The expression of *SUT* gene in transgenic plant LoSUT2 was higher than that of transgenic plant LoSUT4 and the expression of *FT* gene in transgenic plant LoSUT2 was also higher than that of transgenic plant LoSUT4. However, LoSUT2 overexpressing transgenic plants bloomed slightly later than LoSUT4 overexpressing transgenic plants. Researchers have suggested that plant sugar levels are too high to delay flowering and *SOC1* mRNA transcription levels [46]. It was speculated that the functions of different kinds of *SUT* genes in lily were similar but not exactly the same, like *SUT* genes in other plants [47]. LoSUT4 may have a receptive function, which could adjust and control the sugar content of plants, and thus more accurately regulate the expression of flower genes. LoSUT2 may not have this function, resulting in too high a level of plant sugar content that inhibits flowering. Both of them need to work together to regulate the sugar content in tissues of lily and affect the expression of hormones, flowers and other genes, so as to regulate the vernalization and growth process of lily bulbs.

## 4. Materials and Methods

### 4.1. Plant Material

Bulbs of oriental hybrid lily ‘Sorbonne’ were purchased from a local nursery in Yunnan Province after harvest. The bulbs with an even size of ca. 20 g were selected and kept at 4 °C for 63 days. Then they were planted at the Beijing Forestry University, Beijing China (116.3°E, 40.0°N) greenhouse from 1 March to 15 July, 2013 under 22–25/17–20 °C day/night temperature and nature light conditions. Twelve samples of oriental hybrid lily ‘Sorbonne’ new shoot-tip of breaking dormancy bulb were collected. The previous seven groups were collected every 7 days when they were vernalized (at 4 °C 49 days). Then 1–2 cm shoots sprouted at the shoot-tip of flower bud differentiation. Lily buds were collected at five stages, the stage before breaking dormancy, breaking dormancy start-up stage, the flower primordial differentiation stage, the stamen primordial differentiation stage, and the perianth primordial differentiation stage, which was group 8–12. They were immediately snap frozen in liquid nitrogen and stored at −80 °C until used for the experiment.

### 4.2. Analysis of Carbohydrate Metabolism Genes and Construction of Gene Co-Expression Network

A differentially expressed gene (DEG) was declared if the associated FDR ≤ 0.05 and |log2 (ratio)| ≥ 1.5 were observed in three pairwise transcriptome comparisons [48]. Then, through protein sequence alignment, the gene fragments with short sequences and low reliability were deleted, and the DEGs meeting the screening requirements were counted. The heat-map of the carbohydrate metabolism genes was generated using *MeV4.9* and clustered by hierarchical clustering (HCL) with default parameters. To further investigate the interactions between the carbohydrate metabolism genes and flower-related genes, the co-expression network methodology was employed. The Pearson correlation coefficient of the three transcriptome RPKM was calculated using *R. version 3.3.1*. The absolute value of the correlation coefficient greater than 0.85 was selected as the threshold. Those pairs of genes whose coefficient was higher than the threshold were saved to construct the co-expression network by using *Cytoscape* [49].

### 4.3. Total RNA Isolation

Total RNA was extracted using RNAisomate RNA Easyspin Isolation System (Aidlab Biotech, Beijng, China) according to the manufacturer’s instructions. All samples had an RNA concentration >160 µg/µL verified by a 2100 Bioanalyzer (Agilent Technologies, San Diego, CA, United States). A total of 60 µg RNA was pooled from all the samples equally for cDNA preparation. Then we synthesized two paired-end libraries using the Genomic Sample Prep kit (Illumina, San Diego, CA, US) according to the instructions. At last, we purified short fragments with the Qubit TM dsDNA HS Assay kit (Invitrogen, Carlsbad, CA, US) and connected with different sequencing adapters. Both libraries had an average insert size of 400 bp and then we sent them to Shanghai Biotechnology Corporation (Shanghai, China) for sequencing by Illumina HiSeq TM 2000.

### 4.4. Isolation of LoSUT2 Full-Length cDNA

Rapid amplification of cDNA ends (RACE) was used to isolate the complete sequence of the Lo*SUT*2. One microgram of mRNA isolated from SAM was converted into 5′-and 3′-RACE-ready cDNAs with the 5′and 3′ CDS primers by the use of the SMART RACE cDNA amplification kit (Clontech, Palo Alto, CA, USA). According to the partial sequence of the Lo*SUT*2 of the EST clone, specific primers Lo*SUT*2 5f (5′-ATGGCCAACCGTGGCCGCCA-3′), Lo*SUT*2 3r (5′-TTACTTTCGAGTCCTTGTCC-3′) were designed for amplification of the 5′ and 3′ ends, respectively. All PCR products were cloned into pGEMT-T Easy vector (Takara, Shiga, Japan). Plasmid DNAs purified from overnight cultures of three independent clones were sequenced for each transformation, and all resulting sequences were aligned with the partial cDNA sequence by use of the GCG program.

### 4.5. Real-Time Quantitative PCR Verification

Total RNA was isolated from the unvernalized, different vernalized stages and different flower bud differentiation stages, as described above. First-strand cDNA synthesis was performed using Superscript II reverse transcriptase (Invitrogen, Carlsbad, CA, USA) according to the manufacturer’s instructions, using 1 µg total RNA and oligo(dT) primers. qRT-PCR was performed using a Rotor-Gene 3000 real-time PCR detection system (Qiagen) using SYBR^®^ qPCR Mix (Toyobo, Tokyo, Japan) according to the manufacturer’s protocol. The primers used in this study were designed with Beacon Designer (Premier, Palo Alto, CA, USA) and are listed in Appendix A. Real-time PCRs was carried out using prepared cDNA (80 µg) with each set of primers and probes and iQ™ SYBR^®^ Green Supermix (Cat. No.170-8882, Bio-Rad, Hercules, CA, USA). The PCR cycling conditions were as follows: 95 °C (30 s), 60 °C (30 s), and 72 °C (15 s). All reactions were performed in biological triplicates. Relative mRNA levels were calculated using the 2^−^^∆∆^^Ct^ method against the internal reference gene *TIP1*, with expression in the CT unvernalized sample used as the internal control.

### 4.6. Protein Physical and Chemical Properties

The web server of ProtParam was used to calculate the physical and chemical properties of proteins [50].

### 4.7. Prediction of Protein Secondary Structure and Post-Translational Modification

PredictProtein was used to predict the secondary structure (web server: https://www.predictprotein.org/) and TMHMM (web server: http://www.cbs.dtu.dk/services/TMHMM/) and Phobius (web server: http://phobius.sbc.su.se/) were used to predict the transmembrane regions of the protein.

SignalP was used to predict the signal peptide (web server: http://www.cbs.dtu.dk/services/SignalP/). NetAcet was used to predict possible acetylation site (web server: http://www.cbs.dtu.dk/services/NetAcet/). NetPhos was used to predict the phosphorylation site (web server: http://www.cbs.dtu.dk/services/NetPhos/).

### 4.8. Multiple Sequence Alignment and Construction of Phylogenetic Tree

The DNA sequences of 86 *SUT* genes of 36 plant species from NCBI nucleotide and Gene database were collected [50]. These genes are: Dicot species, *Alonsoa meridionalis* Am*SUT*1 (accession number AF191025), *Apium graveolens* Ag*SUT*1 (AF063400), Ag*SUT*2A (AF167415), Ag*SUT*3 (DQ286433), *Arabidopsis thaliana* AtSUC1 (At1g71880), AtSUC2 (At1g22710), AtSUC3/*SUT*2 (At2g02860), AtSUC4/*SUT*4 (At1g09960), AtSUC5 (At1g71890), AtSUC6 (At5g43610), AtSUC7 (At1g66570), AtSUC8 (At2g14670), AtSUC9 (At5g06170), *Asarina barclaiana* Ab*SUT*1 (AF191024), *Beta vulgaris* Bv*SUT*1 (X83850), *Brassica oleracea BoSUC1* (AY065839), Bo*SUC2* (AY065840), *Citrus sinensis* Cs*SUT*1 (AY098891), Cs*SUT*2 (AY098894), *Cucumis melo* Cm*SUT*4 (FJ231518), *Datisca glomerate* Dg*SUT*4 (AJ781069), *Daucus carota* Dc*SUT*1A (Y16766), Dc*SUT*2 (Y16768), *Eucommia ulmoides* Eu*SUT*2 (AY946204), *Euphorbia esula* Ee*SUT*1 (AF242307), *Glycine max* Gm*SUT*1 (AJ563364), *Hevea brasiliensis* Hb*SUT*1 (DQ985466), Hb*SUT*2A (DQ985467), Hb*SUT*2B (DQ985465), Hb*SUT*3 (EF067334), Hb*SUT*4 (EF067335), Hb*SUT*5 (EF067333), Hb*SUT*6 (AM492537), *Juglans regia* Jr*SUT*1 (AY504969), *Lotus japonicus* Lj*SUT*4 (AJ538041), *Lycopersicon esculentum* Le*SUT*1 (X82275), Le*SUT*2 (AF166498), Le*SUT*4 (AF176950), *Malus x domestica* Md*SUT*4 (AY445915), *Manihot esculenta* Me*SUT*1 (DQ138374), Me*SUT*2 (DQ138373), Me*SUT*4 (DQ138371), *Nicotiana tabacum* Nt*SUT*1 (X82276), Nt*SUT*3 (AF149981), *Pisum sativum* Ps*SUT*1 (AF109921), PsSUF1 (DQ221698), PsSUF4 (DQ221697), *Plantago major* PmSUC1 (X84379), PmSUC2 (X75764), PmSUC3 (AJ534442), *Phaseolus vulgaris* Pv*SUT*1 (DQ221699), Pv*SUT*3 (DQ221701), PvSUF1 (DQ221700), *Ricinus communis* RcSCR1 (Z31561), RcSUC4 (AY725473), *Solanum tuberosum* St*SUT*1 (X69165), St*SUT*2 (AY291289), St*SUT*4 (AF237780), *Spinacia oleracea* So*SUT*1 (X67125), *Vicia faba* Vf*SUT*1 (Z93774), *Vitis Vinifera* VvSUC11 (AF021808), VvSUC12 (AF021809), VvSUC27 (AF021810), Vv*SUT*2 (AF439321); Monocot species, *Ananas comosus* Ac*SUT*4 (EF460878), *Bambusa oldhamii* Bo*SUT*1 (DQ020217), *Hordeum vulgare* Hv*SUT*1 (AJ272309), Hv*SUT*2 (AJ272308), Oryza sativa Os*SUT*1 (X87819), Os*SUT*2 (AB091672), Os*SUT*3 (AB071809), Os*SUT*4 (AB091673), Os*SUT*5 (AB091674), *Saccharum hybrid* Sh*SUT*1 (AY780256), *Triticum aestivum* Ta*SUT*1A (AF408842), Ta*SUT*1B (AF408843), Ta*SUT*1D (AF408844), *Zea mays* Zm*SUT*1 (AB008464), Zm*SUT*2 (AY639018), and Zm*SUT*4 (AY581895). Besides six *SUT* genes of *Physcomitrella patent* were also included: Pp*SUT*1 (XM_001752913), Pp*SUT*2 (XM_001778945), Pp*SUT*3 (XM_001777404), Pp*SUT*4 (XM_001768246), Pp*SUT*5 (XM_001766929), and Pp*SUT*6 (XM_001777602).

The CLASTALW was used to do the multiple sequences alignment and a Neighbor-Joining method was used to construct this unrooted tree. The FigTree software was used to show the tree figure. The CLASTALW can be accessed via http://www.ebi.ac.uk/Tools/msa/clustalw2/.

The multiple sequences alignment was conducted by ClastalX to show the conserved sequences. Protein sequence were searched for in the NCBI-conserved domain database to find domains, and Cn3D was used to show the 3D structure of the Major Facilitator Superfamily (MFS) domain.

### 4.9. Analysis of Soluble Sugars, Reducing Sugar and Sucrose

From the beginning of the storage in the lily bulbs, accurately the outer scales (From the outside to the inside of the 3 layers) and the internal scales (scales except the outermost and innermost 3-layer scales) were collected as materials. After weighing and mixing, the materials were frozen by liquid nitrogen for 30 min and placed in a −80 °C refrigerator. The contents of soluble sugars, reducing sugar and sucrose, were carried out by analysis on UV spectrophotometer assays [51].

### 4.10. Analysis of Endogenous Hormones

Auxin (IAA), Abscisic acid (ABA) and Gibberellin (GA) were measured by Enzyme-linked Immunosorbent Assay (ELISA). The kit was provided by China Agricultural University.

### 4.11. Arabidopsis Transformation

pBI121 was used as a vector for target genes. Both of them were cut from XbaI and SalI, at 37 °C for 90 min. It was inserted into pBI121. Clones were transformed by heat shock into Competent E. coli (TransGen Biotech, Inc, Beijing, China) according to the manufacturer’s instructions. *Agrobacterium tumefaciens* strain was applied to *Arabidopsis* [50].

## 5. Conclusions

We found that metabolism genes were closely related to flower genes and hormone genes to affect the process of vernalization and development of lily bulbs and *SUT* gene was the key gene in the process. More importantly, overexpression of *LoSUT* gene would affect the expression of *FT*, *SOC1*, and *LFY* gene and lead to premature flowering of *Arabidopsis thaliana*. Our findings highlighted the importance of *LoSUT* gene in the development of Oriental hybrid lily ‘Sorbone’ and revealed the potential signaling function of *LoSUT* gene during flowering.

## Figures and Tables

**Figure 1 ijms-21-03092-f001:**
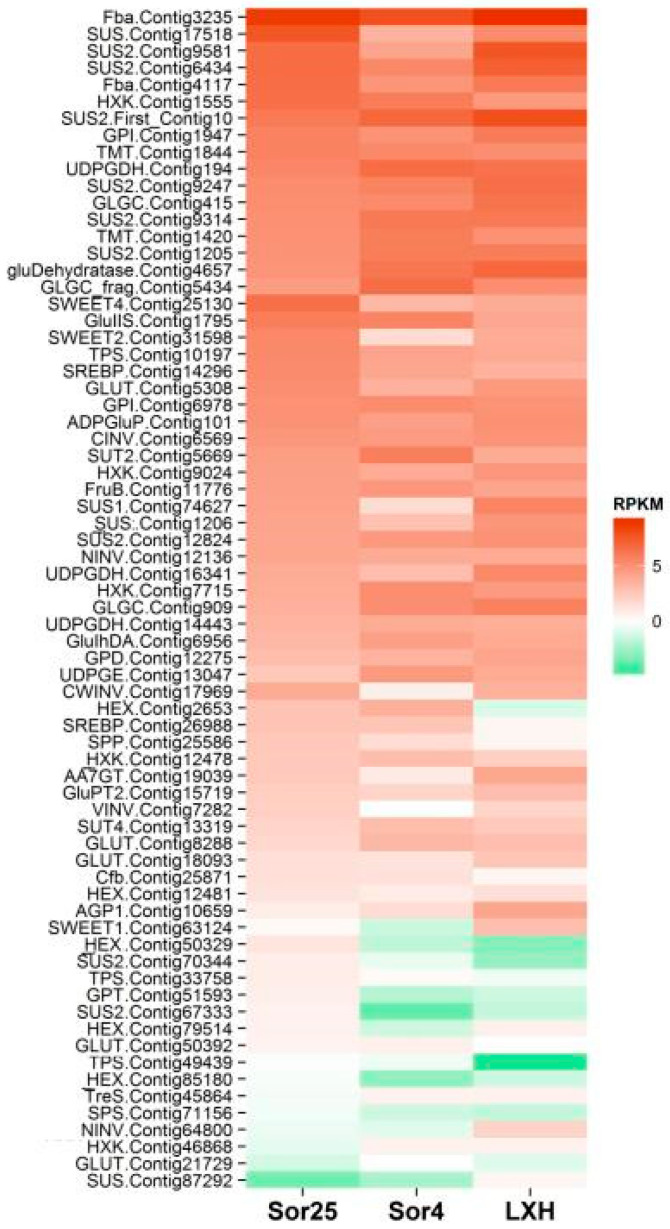
Heatmap of differential gene expression of carbohydrates. In this figure, Sor25 means that the apical meristem of Oriental lily ‘Sorbonne’ is in the period of not vernalized yet, Sor4 means oriental lily ‘Sorbonne’ vernalization completion period in the apical meristem, and LXH means the Oriental lily ‘Sorbonne’ flower bud differentiation period. The expression level represented in the heat map was log2-based. The color scale represents transcript abundance in which green to red represents a change in the expression level from low to high.

**Figure 2 ijms-21-03092-f002:**
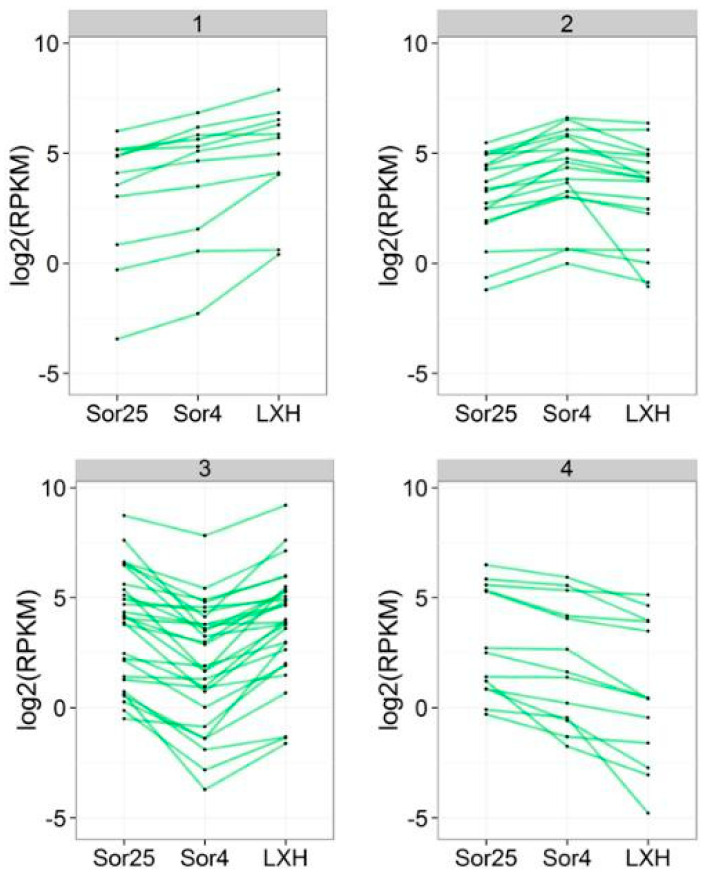
Expression patterns of carbohydrate metabolism genes. Sor25 means that the Oriental lily ‘Sorbonne’ growth point is not vernalized yet, Sor4 means oriental lily ‘Sorbonne’ growth point vernalization completion period, and LXH means the Oriental lily ‘Sorbonne’ growth point flower bud differentiation period.

**Figure 3 ijms-21-03092-f003:**
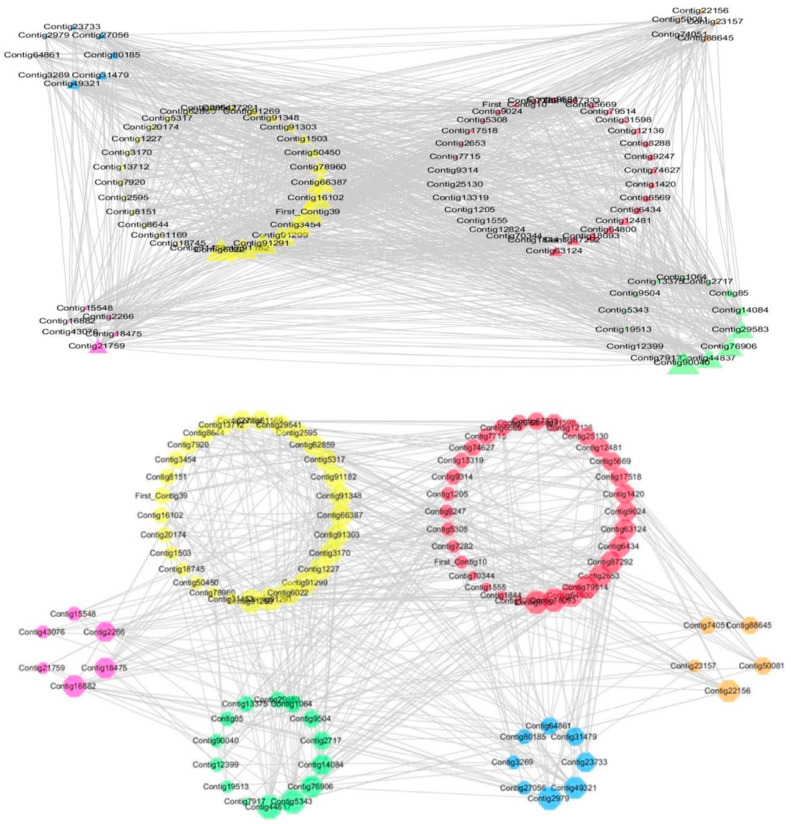
GCN of carbohydrate metabolism genes and flower differentiation, auxin, gibberellin, cold, and methylation-related genes. Different colored nodes represent different types of genes and the edges represent pairwise relations between genes. Yellow nodes: flower differentiation genes; red nodes: sugar transport genes; purple nodes: Gibberellin genes; green nodes: Auxin genes; blue nodes: methylation genes; orange nodes: cold genes. The correlation coefficient of the figure above was 0.95 and the figure below was 0.95.

**Figure 4 ijms-21-03092-f004:**
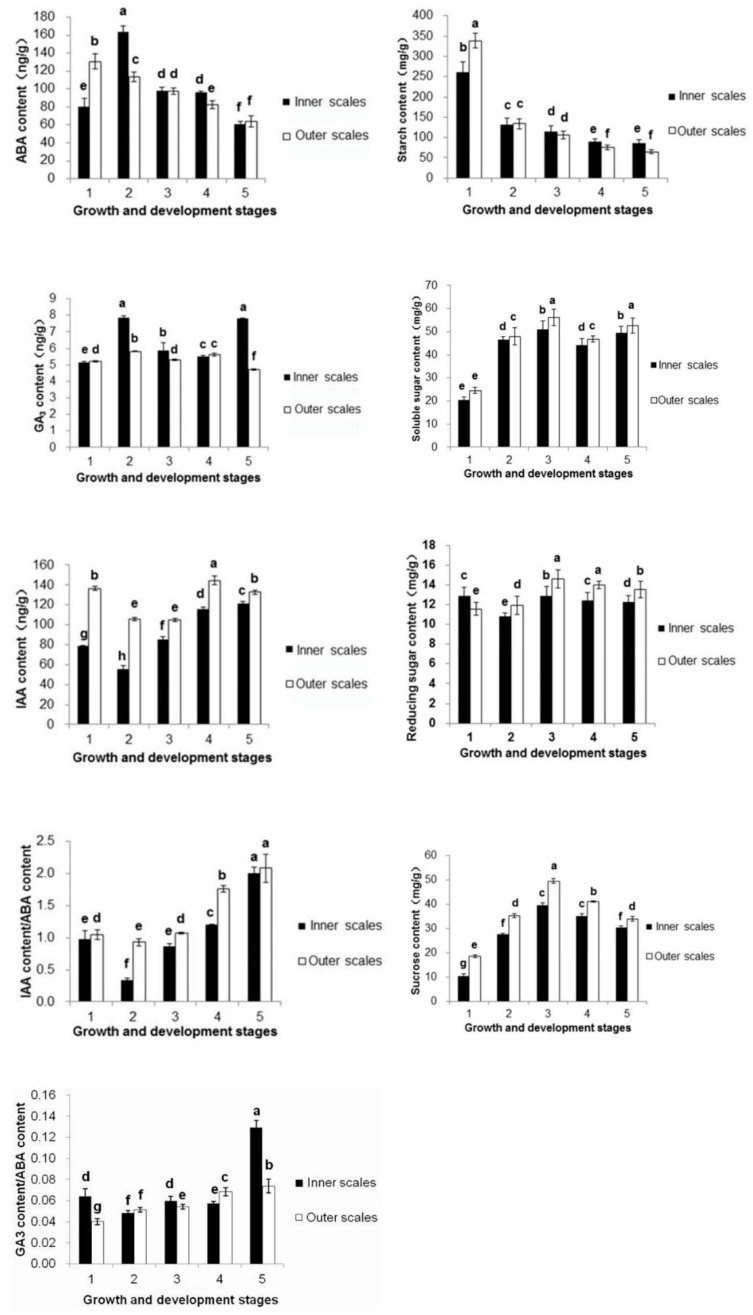
Development of Lily bulbs were divided into five major periods, before vernalization, vernalization, after vernalization, flower bud differentiation, and flower bud development. In order to explore the role of carbohydrate metabolism, changes of carbohydrates and hormones contents in lily bulbs were measured at the physiological level.

**Figure 5 ijms-21-03092-f005:**
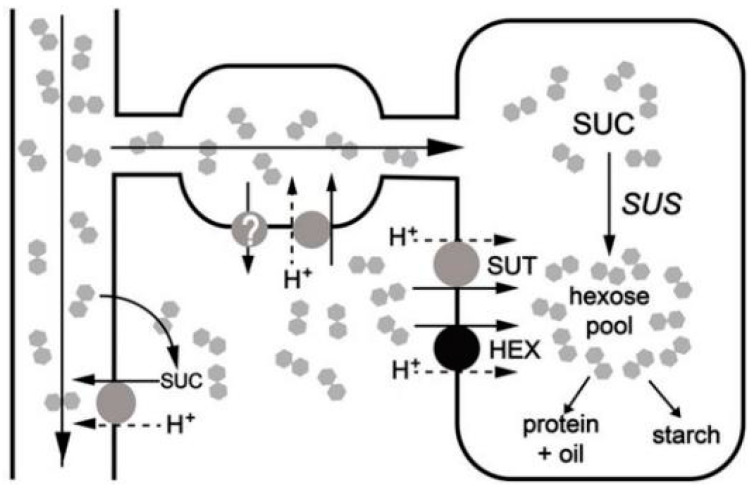
Sucrose transportation in sink tissue [9].

**Figure 6 ijms-21-03092-f006:**
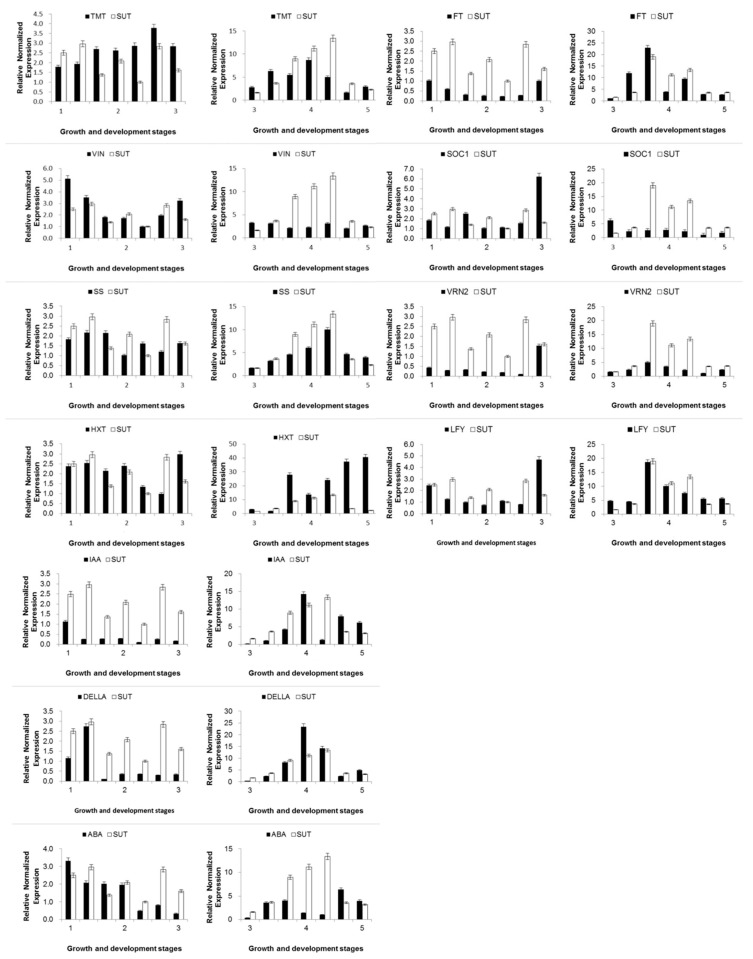
The expression of carbohydrate-related genes, flower differentiation genes and hormone genes in Oriental hybrid lily by qRT-PCR. The relative expression level of the enzyme was calculated to determine the expression level of the gene by 2^−^^∆∆^*^C^*^t^. The instrument was a Bio-rad CFX 96 real-time PCR instrument.

**Figure 7 ijms-21-03092-f007:**
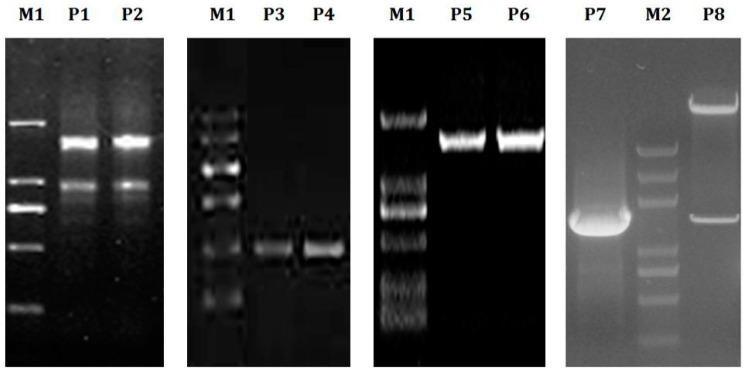
*LoSUT* clone electrophoresis gel map. M1: marker1: The molecular weights in order from large to small: 2000, 1000, 750, 500, 250, 100 bp; M2: marker2 The molecular weights in order from large to small: 5000, 3000, 2000, 1000, 750, 500, 250, 100 bp; P1, P2: RNA Bands; P3, P4: SUT fragment clone; P5, P6: SUT full length cloning; P7: *SUT* gene bacteria PCR; P8: BamHI digestion of expression vector pBI121 through XbaI.

**Figure 8 ijms-21-03092-f008:**
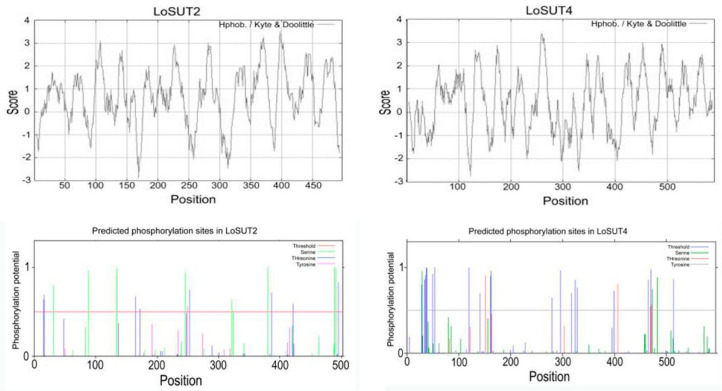
Hydrophobic cluster and signal peptide analysis of LoSUT2 and LoSUT4. The amino acid sequences were arranged horizontally. The hydrophobic site was located above the zero line, while the hydrophilic site was located below the zero line. The horizontal axis represents the amino acid, and the vertical axis represents the probability that the amino acid of that position was phosphorylated.

**Figure 9 ijms-21-03092-f009:**
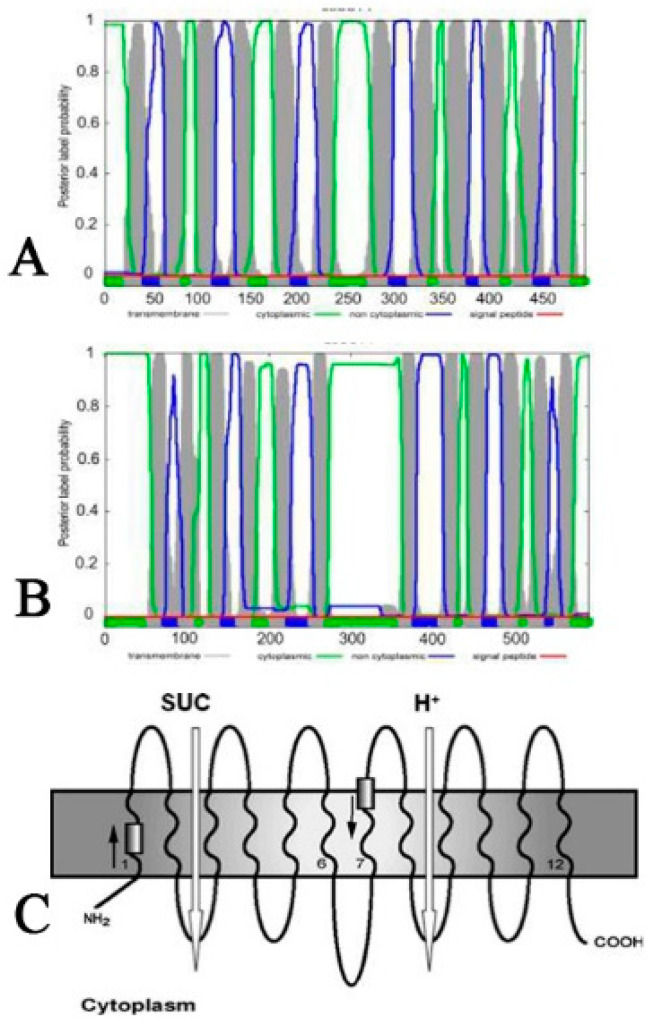
Two-dimensional structural models of protein LoSUT2 and LoSUT4. Among them, Figure 9A is the prediction of the transmembrane region of the LoSUT2 protein; Figure 9B is the predicted result of the transmembrane region of the LoSUT4 protein; and Figure 9C is the classical transmembrane structure of the *SUT* gene summarized in the reference.

**Figure 10 ijms-21-03092-f010:**
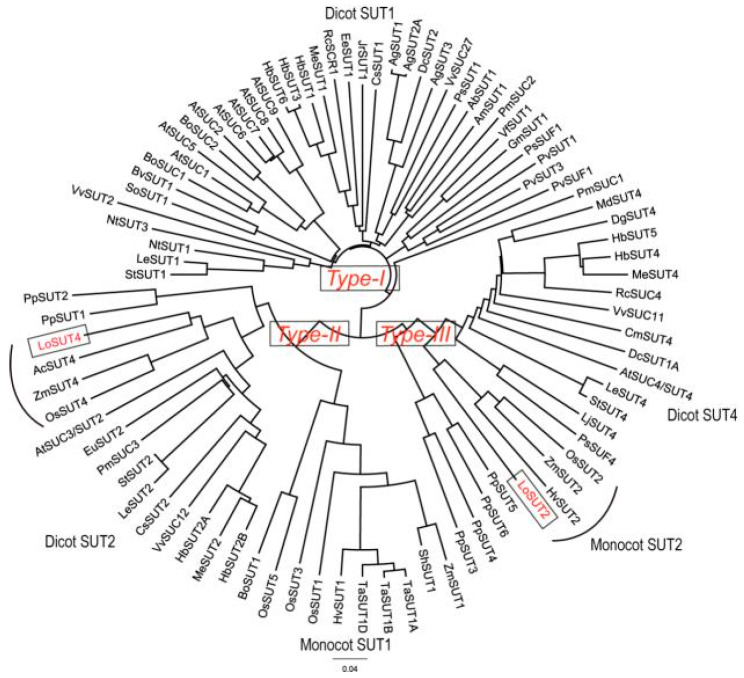
Phylogenetic tree analysis of LoSUT2 and LoSUT4. The multi-sequence alignment and construction of the phylogenetic tree was done by using CLASTALW. The algorithm for constructing the phylogenetic tree was the NJ method. The phylogenetic tree was displayed using FigTree software.

**Figure 11 ijms-21-03092-f011:**
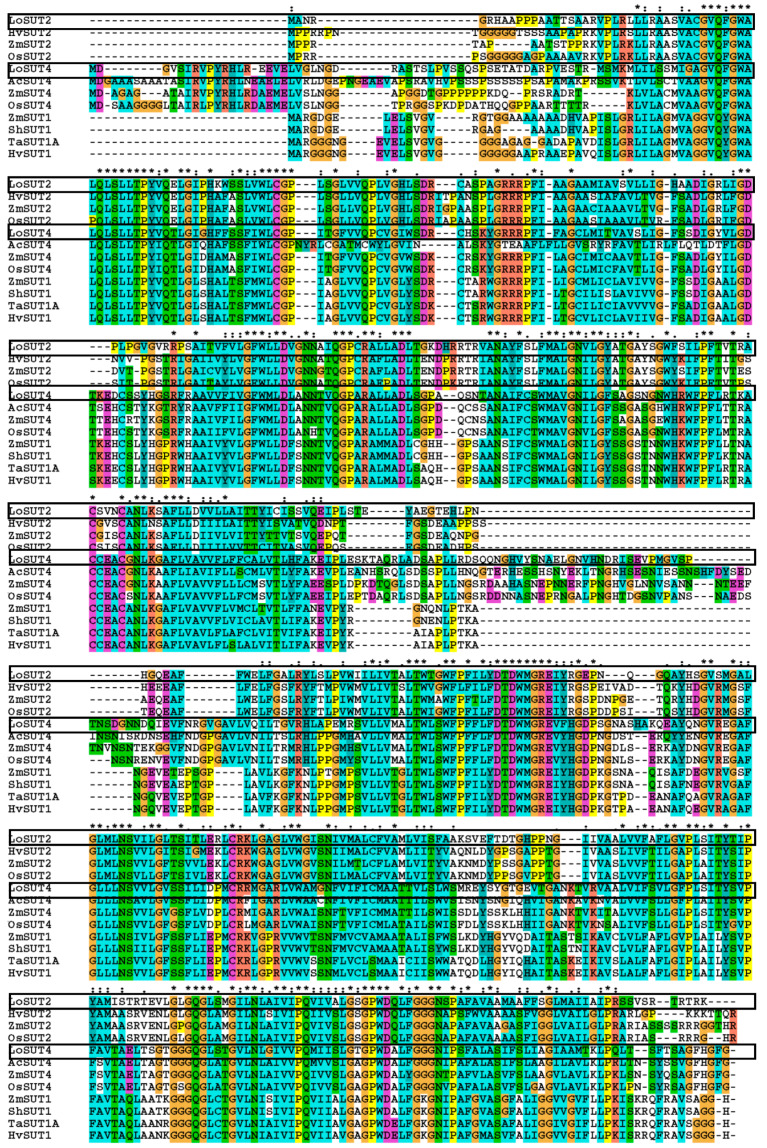
Comparison of the deduced amino acid sequence of *LoSUT2* and *LoSUT4* with its homologs. The black box represents the location of *LoSUT* genes. Amino acid residues of different kinds are shaded in different colors to show the similarity of the sequences.

**Figure 12 ijms-21-03092-f012:**
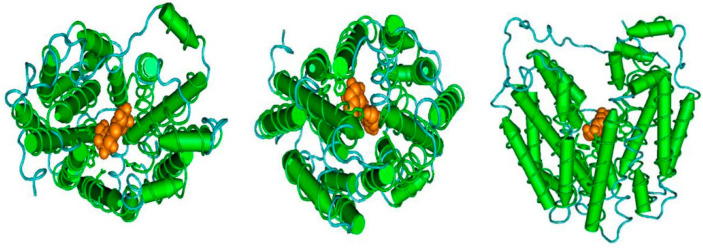
Three-dimensional structural models of protein LoSUT2 and LoSUT4. The software Cn3D was used to show three-dimensional structural models of Major Facilitator Superfamily domain (MFS).

**Figure 13 ijms-21-03092-f013:**
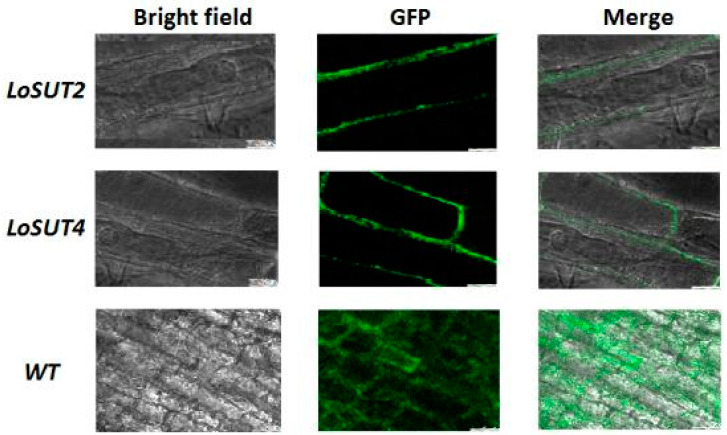
Subcellular localization of *LoSUT2* and *LoSUT4* in onion epidermis. *SUT* genes expressed the GFP-highlighted position of proteins.

**Figure 14 ijms-21-03092-f014:**
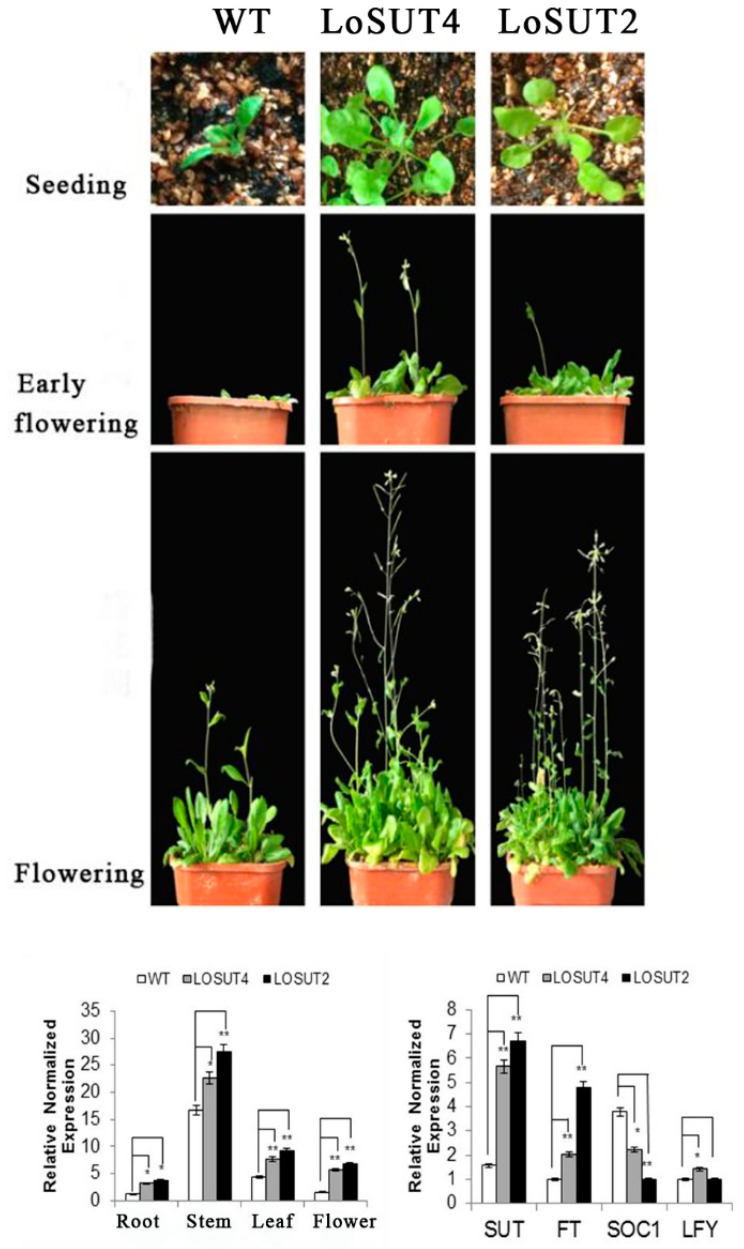
Overexpression of *LoSUT2* and *LoSUT4* in Arabidopsis and qPCR of wide type and transgenic *Arabidopsis thaliana.*

**Table 1 ijms-21-03092-t001:** *INV* genes evaluation form.

Contig	All	All	Cell Wall	Acid	Vacuolar	Cytosolic	Neutral Alkaline	Neutral	
Contig88348	198|334	386	Yes	Yes	Yes		Yes		CW, acid, Missing the front fragment
Contig28073	197|334	237	Yes	Yes	Yes		Yes		*CWINV*, Missing the back fragment
Contig87326	186|334	140	Yes	Yes	Yes		Yes		Not Suitable ORF
Contig31833	156|334	100	Yes	Yes	Yes		Yes		short fragment
Contig1581	135|334	538				Yes	Yes	Yes	NA*INV*, full length
Contig8633	134|334	554				Yes	Yes	Yes	Missing the front fragment, NA*INV*
Contig11809	134|334	543				Yes	Yes	Yes	NA*INV*, full length
Contig6425	134|334	542				Yes	Yes	Yes	NA*INV*, full length
Contig17492	134|334	546				Yes	Yes	Yes	N*INV*, full length
Contig16491	134|334	498				Yes	Yes	Yes	N*INV*, full length
Contig28979	129|334	106				Yes	Yes	Yes	NA*INV*, Only one small fragment later
Contig84458	121|334	75				Yes	Yes	Yes	Not Suitable ORF
Contig43251	107|334	58				Yes	Yes	Yes	Not Suitable ORF

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
