# Peer review of "Transcriptome Analysis of Carbohydrate Metabolism Genes and Molecular Regulation of Sucrose Transport Gene LoSUT on the Flowering Process of Developing Oriental Hybrid Lily ‘Sorbonne’ Bulb"

_ijms, 2020, doi:10.3390/ijms21093092_

Round 1
Reviewer 1 Report
This manuscript presents an interesting study on the analysis of the molecular regulation of sucrose transporter effect on the flowering process of Oriental Hybrid Lily ‘Sorbonne’Bulb.
The subject is of interest for this species of economic importance and for its development. This manuscript corresponds to the standards of the “International Journal of Molecular Sciences. "
The study used various molecular tools, genetic transformation and physiological commonly used.
The manuscript, even if it remains descriptive and uses many abbreviations, the objectives are clear and the conclusion and the abstract reflect the work carried out.
There are, however, many debatable points
From a methodological point of view, and in general, the conditions of growth and development of any plant, have a considerable influence on the visible (growth) and non-visible physiological, genetic and epigenetic processes (transport of metabolites and expression and regulation of genes). This point is not addressed or emphasized in this manuscript. Development conditions are not even displayed. It is a major shortcoming that influences the discussion of the results. This discussion deserves to be improved with more recent references.
From a background point of view.
The authors have limited themselves to references that are old dated.
For example, some recent articles studying several species of Lilium.
Jian-hua Huang, Rong-rong Zhou, Dan He, Lin Chen, Lu-qi Huang. Rapid identification of Lilium species and polysaccharide contents based on near infrared spectroscopy and weighted partial least square method. International Journal of Biological MacromoleculesVolume 1541 July 2020Pages 182-187.
Pengyu Wang, Jian Li, Fatma Alzahra K. Attia, Wenyi Kang, Changqin Li. A critical review on chemical constituents and pharmacological effects of Lilium. Food Science and Human WellnessVolume 8, Issue 4December 2019Pages 330-336.
Farhat Abbas, Yanguo Ke, Yiwei Zhou, Umair Ashraf, Yanping Fan. Molecular cloning, characterization and expression analysis of LoTPS2 and LoTPS4 involved in floral scent formation in oriental hybrid Lilium variety ‘Siberia’. PhytochemistryVolume 173May 2020Article 112294
Shengli Song, Zhiping Wang, Yamin Ren and Hongmei Sun. Full-Length Transcriptome Analysis of the ABCB, PIN / PIN-LIKES, and AUX / LAX Families Involved in Somatic Embryogenesis of Lilium pumilum DC. Fisch. Int. J. Mol. Sci. 2020, 21 (2), 453; https://doi.org/10.3390/ijms21020453 - 10 Jan 2020
Additional remark
Follow the recommendations to authors for indexing bibliographic references
Author Response
Dear reviewer:
On behalf of my co-authors, we here express our sincere gratitude for your valuable comments on our manuscript ijms-769547. We all think that these comments are very valuable and helpful for revising and improving our paper, as well as the important guiding significance to our research. We have studied comments carefully and have made corrections which we hope meet with approval. The revised parts are marked with revision mode in the manuscripts.
The main corrections in the paper and responds to Reviewer 1 comments and suggestions are listed as following.
Comment 1: From a methodological point of view, and in general, the conditions of growth and development of any plant, have a considerable influence on the visible (growth) and non-visible physiological, genetic and epigenetic processes (transport of metabolites and expression and regulation of genes). This point is not addressed or emphasized in this manuscript. Development conditions are not even displayed. It is a major shortcoming that influences the discussion of the results. This discussion deserves to be improved with more recent references.
Response: Thanks for your valuable comments. In this manuscript, we mainly focused on the role of sucrose signaling during the bulb vernalization and flower bud differentiation in lily and the genes co-expression network between SUT gene and carbohydrate metabolic related genes and flower related genes. The result showed that the transgenic plant of SUT gene flowered earlier than the wild type, which suggesting the signal function of sucrose. We all do think that the conditions of growth and development of plants have a considerable influence on the visible and non-visible processes and we really appreciate for your valuable suggestions. They are absolutely helpful for us to further study the development of lily and other plants. In spite of the importance of figure out the development mechanism, it is not our main purpose in the present study. Therefore, after comprehensive consideration and carefully discussion , we decide not to put this in our discussion of results.
Comment 2: From a background point of view. The authors have limited themselves to references that are old dated.
Response: Thanks for your valuable comments. We have referred to the latest references and some more recent papers have been cited. And the bibliographic index of references has been established followed the recommendations to authors.
The above is our responds to the reviewers’ main comments. In all, both two reviewer’s comments are quite helpful and we have revised our paper point-by-point. We have made some changes in the manuscript, which will not influence the content and framework of the paper. We appreciate for Editors and Reviewers’ warm work earnestly and hope the revision will meet with approval. Once again, thank you very much for your comments.
Looking forward to hearing from you.
Thank you and best regards.
Yours sincerely,
Yingmin Lyu and Zheng Zhen

Reviewer 2 Report
In this paper, Gu et al aimed to study the details of the molecular regulation of sugar transporter genes during vernalization and differentiation of flower bud in Lily. However, I found several issues related to technical terms, English language, and typos in the current version. Therefore, I recommend the authors to have their manuscript edited for language by a native English speaker or professional editing service, before resubmit it for peer review.
Please see some of my comments below:
Abstract
- Spell out SUT in the abstract and in the first mention.
Introduction
- Line 39: this sentence should be revised. I feel like the word “most scientist” should not be used. Also, there is only one reference cited. Was this discussed in the cited paper? The paper was published in 2006, therefore, I think the authors should cite some more recent papers, if it is still the case.
- Define sucrose transport (SUT) here in its first mention. SUT was not defined.
- Throughout the introduction: please correct the typos. I think this needs English language editing before sending out for review.
Results:
- I think the authors should start the Results section with some introduction. There was nothing about “annotation”, and the authors just started it with “after annotation”, which I do not think sufficient enough to understand the context of this study.
- Line 88: what are “step” here? I assume that this refers to the step described in the Methods. As I mentioned in the previous comment, the authors should summarize or introduce the readers to what they present here. This is very sudden, in my opinion.
- Line 89: spell out INV in the first mention.
- Line 124: what are “three sequencing periods”? Are they growth stages?
- Line 125 (and figure 2 caption): which R packages? This probably can be presented in the Methods.
Other sections:
- Please check typos, English language and technical terms. There are too many issues with the writing in this manuscript.
Author Response
Dear reviewer:
On behalf of my co-authors, we here express our sincere gratitude for your valuable comments on our manuscript ijms-769547. We all think that these comments are very valuable and helpful for revising and improving our paper, as well as the important guiding significance to our research. We have studied comments carefully and have made corrections which we hope meet with approval. The revised parts are marked with revision mode in the manuscripts.
The main corrections in the paper and responds to Reviewer 2 comments and suggestions are listed as following.
Comment 1: Abstract Spell out SUT in the abstract and in the first mention.
Response: Thanks for your valuable comments. SUT (sucrose transporter) has been spelled out in the abstract and in the first mention.
Comment 2: Line 39: this sentence should be revised. I feel like the word “most scientist” should not be used. Also, there is only one reference cited. Was this discussed in the cited paper? The paper was published in 2006, therefore, I think the authors should cite some more recent papers, if it is still the case.
Response: Thanks for your valuable comments. We all think this statement should be improved and this sentence has been deleted. We further learned about the signal function of sucrose from more recent papers and discussed this question in the Discussion.
Comment 3: Define sucrose transport (SUT) here in its first mention. SUT was not defined.
Response: Thanks for your valuable comments. SUT has been defined in its first mention.
Comment 4: Throughout the introduction: please correct the typos. I think this needs English language editing before sending out for review.
Response: Thanks for your thoughtful comment. We have checked our manuscript and corrected the typos carefully.
Comment 5: I think the authors should start the Results section with some introduction. There was nothing about “annotation”, and the authors just started it with “after annotation”, which I do not think sufficient enough to understand the context of this study.
Response: Thanks for your valuable comments. We have added introductions before we started the Result section and explained the steps for annotation. We all think that it was not sufficient enough for readers to understand this context and we will revise it carefully according to your suggestion in our further study.
Comment 6: Line 88: what are “step” here? I assume that this refers to the step described in the Methods. As I mentioned in the previous comment, the authors should summarize or introduce the readers to what they present here. This is very sudden, in my opinion.
Response: Thanks for your valuable comments. “Step” means the procedures that represent the reannotation of carbohydrate metabolic genes. It has been described in detail in the Methods. We all think this statement was very confusing and very sudden as “annotation” that you mentioned in your previous moment. We have revised it carefully according to your suggestion and we will pay more attention of this problem in our further study.
Comment 7: Line 89: spell out INV in the first mention
Response: Thanks for your valuable comments. We have spelled out INV as well as other acronyms in their first mention.
Comment 8: Line 124: what are “three sequencing periods”? Are they growth stages?
Response: Thanks for your valuable comments. “Three sequencing periods” represents three growth stages of lily bulbs which are the period of not vernalized yet, the period of vernalization completion and the period of flower bud differentiation. It has been explained in the header of Fig 2.
Comment 9: Line 125 (and figure 2 caption): which R packages? This probably can be presented in the Methods.
Response: Thanks for your valuable comments. The R version 3.3.1 was used to analyze data and construct the gene co-expression network. And it has been presented in the Methods according to your comment.
The above is our responds to the reviewers’ main comments. In all, both two reviewer’s comments are quite helpful and we have revised our paper point-by-point. We have made some changes in the manuscript, which will not influence the content and framework of the paper. We appreciate for Editors and Reviewers’ warm work earnestly and hope the revision will meet with approval. Once again, thank you very much for your comments.
Looking forward to hearing from you.
Thank you and best regards.
Yours sincerely,
Yingmin Lyu and Zheng Zhen

Round 2
Reviewer 1 Report
Manuscript was improved as requested, the modifications have been made to remove any ambiguity and give clarity to the approach implemented for this study.
Author Response
Thank you very much for your reviewing.
Reviewer 2 Report
In general, I think the manuscript is technically sound, but I cannot help the feeling that this manuscript needs extensive English editing to be more concise and comprehensive, before it can be published. There are many vague and confusing expression throughout the manuscript.
Additionally, while the results look good, the authors tried to include too much of them in the main text. The focus of this study is, as stated in the title, sucrose transporters (SUTs), so I think some less related results (e.g., in sections 21-2.5) can be presented as Supplementary documents. Other option could be to change the title to include “carbohydrate metabolism genes”, instead of SUTs. Overall, I think it is important to decide which results (from your good data) should be in the main text, and which should be in the supplementary.
Please see my comments below:
Title:
- I think, many cases, the title is the only information that the readers see first before deciding to read your paper further, so I would suggest the authors to use “sugar transporters” instead of SUT.
Abstract:
- "third structure", change to tertiary structure
Introduction:
- Line 37: it needs
- Line 41: transported from sink to source tissues instead of “translated”?
- Line 62: consider replacing “always” by “mostly”.
- Line 65: “two or more”, what is exactly the number of genes was reported for monocot species here?
- Line 65: what relevant?
- Line 68: this sentence is a bit confusing to me. Did the authors mean “most studies on molecular mechanism focused on temperature and light, but not carbohydrates?”. The please check the sentence.
Results:
- Consider replacing the word “discovered” by “studied”.
- Line 82: change to re-annotated
- Line 96: Did the authors used any packages to generate the heatmap in R, or what is the function used? Using “constructed in R language” is not sufficient”, in my opinion.
- Table 1: I do not understand the logic of presenting INV genes in this Table 1, while the focus of this study is SUTs. I think, the authors could present this Table as a whole for all 25 genes that they identified, in the Supplementary, including the full-names for all genes that were presented in Figure 1. I think it is important to include detailed data related to the 25 genes that the authors started with. The current Table 1 can be placed in the Supplementary document.
- Line 122: again, related to R language, which package or function for clustering, specifically?
- Line 132: model or mode?
- Would be better to use "cluster" or "gene cluster", instead of “mode”, since it was resulted from clustering.
- Paragraph line 151: this is very confusing part.
- Figure 3 caption: what are “the above genes’? What do the lines (that gene are connected with) represent here? Correlation coefficient? This should be included in the caption.
- Line 234: at higher levels?
- Line 375: is “a long A” poly A?
- Line 405: change to Columbia
- Paragraph 414: probably good to use either WT or non-transgenic plant consistently.
Author Response
Dear reviewer:
On behalf of my co-authors, we here express our sincere gratitude for your valuable comments on our manuscript ijms-769547. We all think that these comments are very valuable and helpful for revising and improving our paper, as well as the important guiding significance to our research. We have studied comments carefully and have made corrections which we hope meet with approval. The revised parts are marked with revision mode in the manuscripts.
The main corrections in the paper and responds to Reviewer 2 comments and suggestions are listed as following.
In general, I think the manuscript is technically sound, but I cannot help the feeling that this manuscript needs extensive English editing to be more concise and comprehensive, before it can be published. There are many vague and confusing expression throughout the manuscript.
Additionally, while the results look good, the authors tried to include too much of them in the main text. The focus of this study is, as stated in the title, sucrose transporters (SUTs), so I think some less related results (e.g., in sections 21-2.5) can be presented as Supplementary documents. Other option could be to change the title to include “carbohydrate metabolism genes”, instead of SUTs. Overall, I think it is important to decide which results (from your good data) should be in the main text, and which should be in the supplementary.
Comment 1: Title: I think, many cases, the title is the only information that the readers see first before deciding to read your paper further, so I would suggest the authors to use “sugar transporters” instead of SUT.
Response: Thanks for your valuable comments. As your kindly comments mentioned above, the tile did not cover the main content of this paper completely so that the focus of this text was confusing and the readers would be confused when they saw ‘SUT’ for the first sight. In this paper, our goal was to figure out the function of carbohydrate metabolism genes in the vernalization and flower development of oriental hybrid lily ‘Sorbonne’ bulb through the transcriptome analysis and find out the key gene. Thus both analysis of carbohydrate metabolism genes and identification of the key gene SUT are of equal importance. According to your suggestions, we have revised the title to Transcriptome Analysis of Carbohydrate Metabolism Genes and Molecular Regulation of Sucrose Transport Gene LoSUT on the Flowering Process of Developing Oriental Hybrid Lily ‘Sorbonne’ Bulb. We sincerely hope this title could meet with your approval.
Comment 2: Abstract: "third structure", change to tertiary structure
Response: Thanks for your valuable comments. We have changed ‘third structure’ to ‘tertiary structure’ (line 29).
Comment 3: Introduction: Line 37: it needs
Response: Thanks for your valuable comments. We have already corrected these grammatical problems (line 41-42).
Comment 4: Introduction: Line 41: transported from sink to source tissues instead of “translated”?
Response: Thanks for your valuable comments. The word ‘translated’ has been changed to ‘transported’ (line 45).
Comment 5: Introduction: Line 62: consider replacing “always” by “mostly”.
Response: Thanks for your valuable comments. We have corrected this not rigorous expression in our manuscript and replaced ‘always’ by ‘mostly’ (line 70).
Comment 6: Introduction: “two or more”, what is exactly the number of genes was reported for monocot species here?
Response: Thanks for your thoughtful comment. In this sentence, it means that for most dicot plants, two or more kinds of SUT genes in each species have been reported. For different species, the numbers are not the same, like we have described in this manuscript after this sentence that nine kinds of SUTs were identified in Arabidopsis and three were reported in Pyrus (line 72-75). It is not available to know the exact number of genes in dicot plants so we illustrated the number of SUT genes in certain species.
Comment 7: Introduction: Line 65: what relevant?
Response: Thanks for your valuable comments. We do think this sentence was not clearly expressed. ‘Relevant’ means studies on sucrose transporter in monocot plants. We have changed this sentence to “In contrast to numerous studies on sucrose transport in dicotyledons, the function of SUT genes of many monocotyledons remains largely unknown” (line 76-77).
Comment 8: Introduction: Line 68: this sentence is a bit confusing to me. Did the authors mean “most studies on molecular mechanism focused on temperature and light, but not carbohydrates?”. The please check the sentence.
Response: Thanks for your valuable comments. This sentence means that most studies on the mechanism of lily bulb vernalization focused on temperature, light, hormones etc., and there is less focus on the metabolism of carbohydrate in the process, even if there is, are mainly concentrated on physiology like determination of sugar content, while the molecular mechanism needs to be further studied (line 79-84). We have revised it carefully according to your suggestion and we will pay more attention to our written English in our further study.
Comment 9: Results: Consider replacing the word “discovered” by “studied”.
Response: Thanks for your valuable comments. The word ‘discovered’ has been replaced by ‘studied’ according to your comment (line 92).
Comment 10: Results: Line 82: change to re-annotated
Response: Thanks for your valuable comments. We have corrected this typo (line 95) and we will be more careful about typo issues in our further study.
Comment 11: Results: Line 96: Did the authors used any packages to generate the heat-map in R, or what is the function used? Using “constructed in R language” is not sufficient”, in my opinion.
Response: Thanks for your valuable comments. The heat-map of gene expression was log2-based (line 123) and was generated using MeV4.9, which has been added to Methods (line 600).
Comment 12: Results: Table 1: I do not understand the logic of presenting INV genes in this Table 1, while the focus of this study is SUTs. I think, the authors could present this Table as a whole for all 25 genes that they identified, in the Supplementary, including the full-names for all genes that were presented in Figure 1. I think it is important to include detailed data related to the 25 genes that the authors started with. The current Table 1 can be placed in the Supplementary document.
Response: Thanks for your valuable comments. As we have discussed in Comment 1, there are two focus in this paper, one is to figure out the function of carbohydrate metabolism genes in vernalization and the other one is to identify the molecular regulation of the key gene (SUT) flowering development of lily bulb. Since we have changed the title, we think it makes sense to represent INV genes as an example. Moreover, the INV genes, of which had both sequences not meeting the requirements and fully compliant to the requirements (line 101) could represent the gene identification process better. In addition, according to you comment, the detailed data related to the 70 sequences were listed. As some information (like gene name and contig ID) of the 70 sequences has already represented in Figure 1 and there are too much pages in the table, the detailed data related to the genes are placed in the Supplementary Table 1 but not in the main text.
Comment 13: Results: Line 122: again, related to R language, which package or function for clustering, specifically?
Response: Thanks for your valuable comments. The genes were clustered by hierarchical clustering (HCL) with default parameters, which has been added to Methods (line 601).
Comment 14: Results: Line 132: model or mode? Would be better to use "cluster" or "gene cluster", instead of “mode”, since it was resulted from clustering.
Response: Thanks for your valuable comments. Since the expression patterns were constructed by clustering, ‘Cluster’ has taken place of ‘Mode’ according to your comment (line 141).
Comment 15: Results: Paragraph line 151: this is very confusing part. Figure 3 caption: what are “the above genes’? What do the lines (that gene are connected with) represent here? Correlation coefficient? This should be included in the caption.
Response: Thanks for your valuable comments. In part 2.3, ‘the above genes’ are carbohydrate metabolism genes and flower differentiation, auxin, gibberellin, cold and methylation related genes which were found to have significant differences of expression in the process of vernalization (line 152)and it has been described in detail in the revised version. Besides, the edges and nodes presented and correlation coefficient in Figure 3 are explained in the caption according to your comments (line 193). Different colored nodes represent different types of genes and the edges represent pairwise relations between genes, red nodes: sugar transport genes; purple nodes: Gibberellin genes; green nodes: Auxin genes; blue nodes: methylation genes; orange nodes: cold genes. The correlation coefficients were 0.95 and 0.99, respectively.
Comment 16: Results: Line 234: at higher levels?
Response: Thanks for your kind comments. It has been corrected to higher level (271) and we will be more careful about the grammar issues in our further study.
Comment 17: Results: Line 375: is “a long A” poly A?
Response: Thanks for your valuable comments. We feel so sorry to make this typo mistake and we were going to express ‘a long non-conservative area’ and this has been corrected in our manuscript (line 393).
Comment 18: Results: Line 405: change to Columbia
Response: Thanks for your valuable comments. It has been corrected to ‘Columbia’ (line 426).
Comment 19: Results: Paragraph 414: probably good to use either WT or non-transgenic plant consistently.
Response: Thanks for your valuable comments. We have used WT instead of non-transgenic plant in this manuscript according to your comment (line 433, 436 etc.).
The above is our responds to the reviewers’ main comments. In all, reviewer’s comments are quite helpful and we have revised our paper point-by-point. We have made some changes in the manuscript, which will not influence the content and framework of the paper. We appreciate for Editors and Reviewers’ warm work earnestly and hope the revision will meet with approval. Once again, thank you very much for your comments.
Looking forward to hearing from you. Thank you and best regards.
Yours sincerely,
Yingmin Lyu and Zen Zeng

Round 3
Reviewer 2 Report
The authors have addressed my comments in this version.